# Revised roles of ISL1 in a hES cell-based model of human heart chamber specification

Roberto Quaranta[1,2], Jakob Fell[1,2], Frank Rühle[3], Jyoti Rao[1,2†], Ilaria Piccini[1,2], Marcos J Araúzo-Bravo[4,5], Arie O Verkerk[6,7], Monika Stoll[3,8], Boris Greber[1,2‡*]

[1]Human Stem Cell Pluripotency Laboratory, Max Planck Institute for Molecular Biomedicine, Münster, Germany; [2]Chemical Genomics Centre of the Max Planck Society, Dortmund, Germany; [3]Department of Genetic Epidemiology, Institute of Human Genetics, University of Münster, Münster, Germany; [4]IKERBASQUE, Basque Foundation for Science, Bilbao, Spain; [5]Group of Computational Biology and Systems Biomedicine, Biodonostia Health Research Institute, San Sebastián, Spain; [6]Department of Clinical and Experimental Cardiology, Academic Medical Center, University of Amsterdam, Amsterdam, Netherlands; [7]Department of Medical Biology, Academic Medical Center, University of Amsterdam, Amsterdam, Netherlands; [8]Department of Biochemistry, Cardiovascular Research Institute Maastricht, Maastricht University, Maastricht, Netherlands

*For correspondence: b.greber@rheincell.de

Present address: †Department of Genetics, Harvard Medical School and Brigham and Women's Hospital, Boston, United States; ‡RheinCell Therapeutics GmbH, Langenfeld, Germany

Competing interests: The authors declare that no competing interests exist.

**Abstract** The transcription factor ISL1 is thought to be key for conveying the multipotent and proliferative properties of cardiac precursor cells. Here, we investigate its function upon cardiac induction of human embryonic stem cells. We find that ISL1 does not stabilize the transient cardiac precursor cell state but rather serves to accelerate cardiomyocyte differentiation. Conversely, ISL1 depletion delays cardiac differentiation and respecifies nascent cardiomyocytes from a ventricular to an atrial identity. Mechanistic analyses integrate this unrecognized anti-atrial function of ISL1 with known and newly identified atrial inducers. In this revised view, ISL1 is antagonized by retinoic acid signaling via a novel player, MEIS2. Conversely, ISL1 competes with the retinoic acid pathway for prospective cardiomyocyte fate, which converges on the atrial specifier NR2F1. This study reveals a core regulatory network putatively controlling human heart chamber formation and also bears implications for the subtype-specific production of human cardiomyocytes with enhanced functional properties.

DOI: https://doi.org/10.7554/eLife.31706.001

## Introduction

The four chambers of the mammalian heart are specified from the first and second heart fields (FHF/SHF) encompassing distinct precursor cell populations that give rise, respectively, to the left ventricle (FHF) and the mostly SHF-derived right ventricle, left and right atria, and outflow tract (*Buckingham et al., 2005*). The LIM domain transcription factor ISL1 (Islet-1) is a prime player in and marker of the SHF from which, accordingly, both ventricular and atrial cells originate (*Cai et al., 2003*). The view that the SHF not only gives rise to the right ventricle and outflow tract, but also to most cells of the atria, is a result of revised lineage-tracing experiments in the mouse embryo using improved *Isl1*-Cre driver lines (in *Yang et al., 2006*). Conversely, mouse embryos deficient for *Isl1* show severe cardiac defects highlighting the functional importance of ISL1 in this context (*Cai et al., 2003*).

Proliferating ISL1[+] cells not only form cardiomyocytes (CMs) but they also bear multipotent differentiation potential for generating the smooth muscle and endothelial cell lineages, as shown in vitro and in vivo (*Laugwitz et al., 2008*; *Moretti et al., 2010*, *2006*). Interestingly, human embryonic stem cells (hESCs) as well as induced pluripotent stem cells may give rise to ISL1[+] cells with multipotent properties (*Bu et al., 2009*; *Moretti et al., 2010*). Subsequently, these findings, as well as the fact that ISL1[+] cells undergo significant expansion in vivo (*Cai et al., 2003*), have stimulated efforts to stably propagate them in vitro and potentially pave the way for regenerative medical approaches. ISL1 has thus been used as a main self-renewal marker in these studies (*Cao et al., 2013*; *Cohen et al., 2007*; *Qyang et al., 2007*; *Zhang et al., 2016*).

A prime morphogen playing into SHF development is retinoic acid (RA). It is synthesized by the somites of the mouse embryo, to then signal to the posterior part of the SHF (*Duester, 2008*). At around E7.5, RA restricts the cardiac progenitor pool marked by ISL1, to subsequently promote atrial specification of the posterior SHF. Conversely, in embryos deficient in synthesizing RA, the ISL1-expressing domain of the late SHF, the anterior SHF, is expanded and atrial induction compromised (*Ryckebusch et al., 2008*; *Sirbu et al., 2008*; *Zaffran et al., 2014*). Interestingly, human pluripotent stem cells undergoing cardiac induction are responsive to RA - much like in the in vivo situation. Hence, activation of RA signaling promotes atrial specification at the expense of a default ventricular cell fate (*Ma et al., 2011*). Importantly, Devalla and colleagues have recently shown that NR2F1 (also known as COUP-TFI) is a pivotal RA-induced transcription factor. It activates at least part of an atrial-specific gene expression program including, for instance, the potassium ion channel-encoding *KCNA5* gene mediating atrial-specific action potential properties (*Devalla et al., 2015*; *Marczenke et al., 2017b*).

Moreover, in a recently established differentiation protocol, hESCs homogeneously pass through a transient ISL1 stage before acquiring a terminally differentiated cardiomyocyte (CM) state (*Rao et al., 2016*; *Zhang et al., 2015*). This fact prompted us to functionally investigate the role of ISL1 in hESCs undergoing cardiac differentiation. By combining directed cardiac differentiation of hESCs, targeted genetic manipulation, and functional genomics analysis, we show that ISL1 does not sustain self-renewal of cardiac precursor cells. Rather, it acts as an accelerator of cardiomyocyte differentiation and concurrently takes on a central position in the cardiac subtype specification network. Hence, we find that ISL1 is negatively linked to known and previously unrecognized drivers of atrial induction, NR2F1 and MEIS2, and that it, thereby, acts as a functional opponent of retinoic acid signaling in competing for ventricular versus atrial specification.

## Results

### ISL1 accelerates pan-cardiac gene induction in hESCs without affecting proliferation

Given its key role in vertebrate cardiogenesis and its implication in the cardiac precursor cell state, we sought to investigate the function of ISL1 upon cardiac induction of human ES cells. To this end, a functional knockout-causing deletion was induced in HuES6 cells using CRISPR/Cas9n (*Figure 1A*). Clonal ISL1 knockout (KO) hESCs were then differentiated using a high-efficiency monolayer protocol (*Figure 1B*; *Zhang et al., 2015*). At day 5, the approximate peak expression time point of ISL1 in this protocol (*Rao et al., 2016*), ISL1 was undetectable in KO cells, as expected (*Figure 1C*). Using time course gene expression analysis of various cardiac markers, we noticed with interest that ISL1 KO cells were not entirely deficient in undergoing differentiation into cardiomyocytes. Rather, they displayed a delayed induction of structural as well as regulatory cardiac genes but eventually, they also started to contract spontaneously (*Figure 1D* and *Figure 1—figure supplement 1A*, *Videos 1* and *2*). These results were confirmed at the protein level where wild-type (WT) controls showed robust abundance of cardiac markers by day 6, the usual time point of spontaneous beating initiation in the protocol, whereas ISL1 KO showed a still incomplete pattern by day 8 (*Figure 1E*). Hence, ISL1 is not absolutely required for CM differentiation in hESCs but its depletion slows down the process.

Interestingly, different cardiac genes were affected to different degrees in ISL1 KO cells regarding their induction kinetics. To understand whether this may conversely imply an active role of ISL1 in driving the differentiation process, we generated a stable cell line on ISL1 KO background in

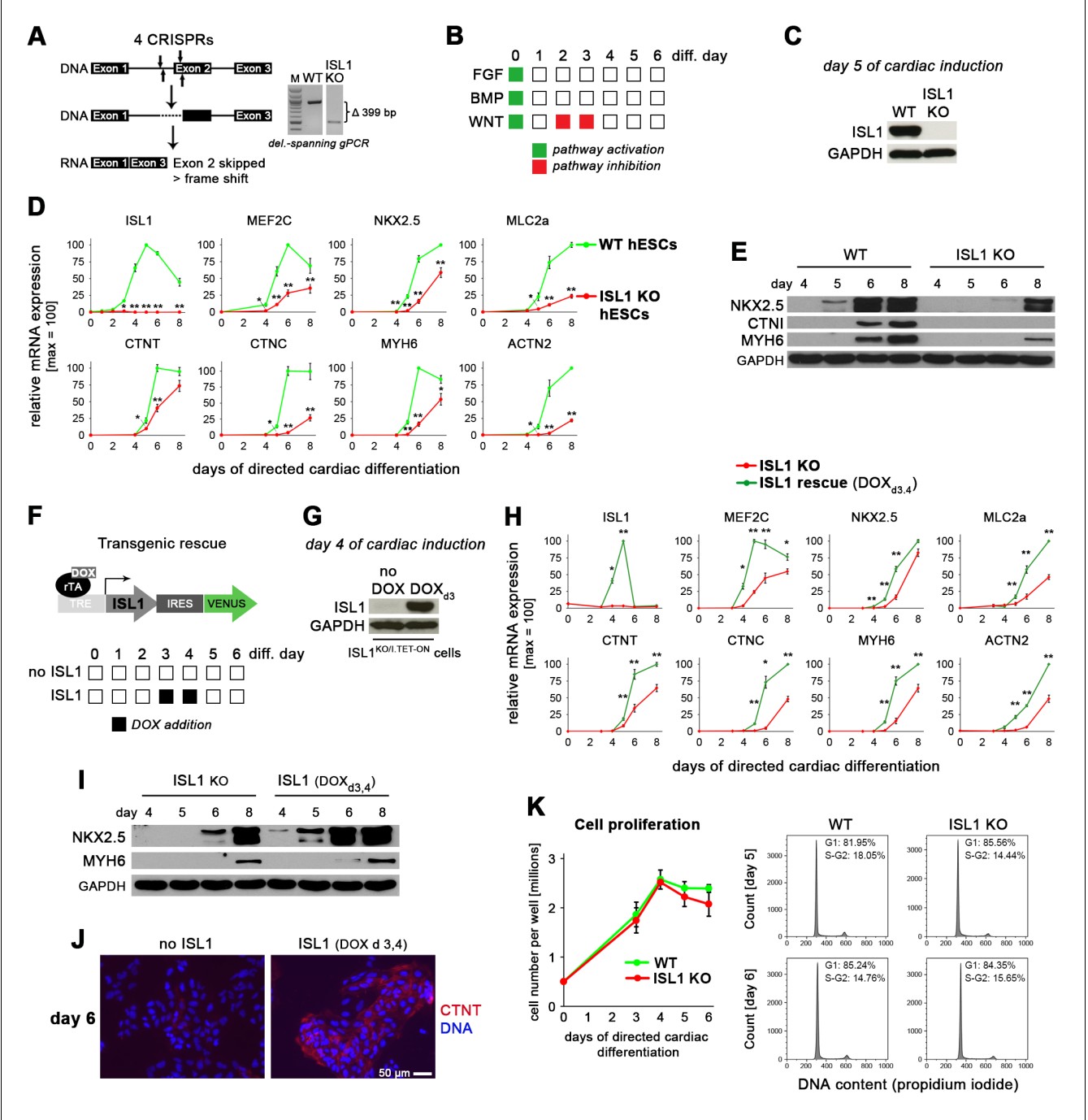

**Figure 1.** ISL1 accelerates cardiac differentiation of hESCs without affecting cell proliferation. (A) CRISPR-mediated knockout of *ISL1* in hESCs. Left: Strategy to delete the intron 1/exon 2 splice junction using two pairs of CRISPR/Cas9 nickase vectors. Right: Validation of induced genomic deletion by genomic PCR in a positive cell line. (B) Schematic of differentiation protocol. The indicated signaling factor treatments underly all cardiac induction experiments throughout this study. (C) Immunoblot validating the absence of ISL1 protein in KO cells at day 5 following the protocol of panel B. (D) Time course gene expression analysis of cardiac genes by RT-qPCR (n = 2–7). (E) Immunoblot confirming the differentiation delay of ISL1 KO cells at the protein level. (F) Schematic of inducible expression vector (top) and doxycycline treatment protocol used in most experiments to mimic the temporal *ISL1* expression pattern of WT cells during directed cardiac induction (bottom). (G) Immunoblot confirming *ISL1* inducibility in clonal ISL1[KO/I.TET-ON] cells during directed cardiac differentiation. (H) Time course gene expression analysis of cardiac genes in ISL1[KO/I.TET-ON] cells (RT-qPCR, n = 2–6). (I) Western blot showing restored kinetics of cardiac differentiation following pulsed *ISL1* induction in ISL1[KO/I.TET-ON] cells. (J) Immunostaining of cardiac troponin T at day 6 of directed differentiation. (K) Cell proliferation measured by cell counting (left, n = 4) and cell cycle analysis (right) of WT and ISL KO hESCs undergoing directed cardiac induction.

DOI: https://doi.org/10.7554/eLife.31706.002

*Figure 1 continued on next page*

*Figure 1 continued*

The following figure supplement is available for figure 1:

**Figure supplement 1.** Delayed cardiac differentiation in ISL1 KO hESCs and rescue using pulsed ISL1 overexpression.
DOI: https://doi.org/10.7554/eLife.31706.003

which an *ISL1* transgene could be selectively induced by doxycycline addition. This strategy would allow to investigate the function of ISL1 in a stage-specific manner, by mimicking the transient induction pattern of endogenous *ISL1*, for instance (*Figure 1F*; compare to ISL1 in *Figure 1D*). The resulting cell line, termed ISL1$^{KO/I.TET-ON}$, showed robust ISL1 expression following a 1-day exposure to DOX (*Figure 1G*). Transient *ISL1* induction in these cells during cardiac differentiation fully restored WT-like gene expression kinetics based on a panel of markers, which also confirmed the specificity of the underlying genetic manipulations (*Figure 1H* and *Figure 1—figure supplement 1C*). Importantly, some of these genes, notably the key cardiac regulators *MEF2C* and *NKX2.5*, showed immediate-early responses to the reintroduction of ISL1 as they became significantly upregulated after only 1 day of DOX addition (*Figure 1H* and *Figure 1—figure supplement 1B–D*). This acceleration effect regarding the upregulation of key cardiac genes also became apparent at the protein level as well as by visual inspection (*Figure 1I* and *Figure 1—figure supplement 1J*, *Video 3*).

To obtain a more global insight, we recorded genome-wide expression time-series using WT and $\pm$ DOX$_{d3,4}$-treated ISL1$^{KO.TET-ON}$ cells (*Supplementary file 1*). Stringent filtering of these data yielded a set of approximately 70 known cardiac and uncharacterized genes, all showing accelerated induction kinetics as driven by ISL1 (*Figure 1—figure supplement 1E–G*). Moreover, cell number as well as cell cycle stage quantification in differentiating WT and ISL1 KO cells revealed only marginal differences between these, which thus could not explain the pronounced differences in cardiac gene induction seen at days 5–6 (*Figure 1K*). These analyses therefore suggest that ISL1 accelerates CM formation in differentiating hESCs by activating downstream pro-cardiac genes, rather than through promoting proliferation or stability of the cardiac precursor cell state, for instance.

## ISL1 does not stabilize the cardiovascular progenitor cell state

To further investigate this latter possibility and to illuminate the putative multipotency of ISL1$^+$ cells, we screened a panel of candidate signaling factors for their ability to either stabilize the precursor state and prevent further differentiation, or to induce different differentiation outcomes, notably cardiomyocytes, smooth muscle, or endothelial cells (*Figure 2A*). Several molecules promoted the specific induction of one or two or these differentiation fates. For example, FGF2 and VEGF clearly stimulated endothelial differentiation from the ISL1$^+$ intermediate stage (*Figure 2—figure supplement 1A–C*). Somewhat surprisingly, however, none of the molecules, including previously reported factors and cocktails, could significantly promote *ISL1* expression which eventually declined under all conditions tested (*Figure 2B and C*).

Next, we challenged the system by applying continuous *ISL1* overexpression in a newly generated WT$^{I.TET-ON}$ line (*Figure 2D*). Sustaining robust ISL1 levels by continuous DOX administration from day 3 of directed cardiac induction onwards did not interfere with CM formation and did not further accelerate the differentiation process (*Figure 2D and E*). Somewhat surprisingly, these data imply that ISL1 plays no role in sustaining the cardiac precursor cell state. As a confirmation, *ISL1* activation in ISL1$^{KO/I.TET-ON}$ cells did not affect the expression of independent precursor genes like *PDGFRA* (*Figure 2—figure supplement 1D*). We did, however, notice that a later induction of *ISL1* in already-formed CMs specifically antagonized the expression of ventricular-specific genes, notably that of *MLC2v*, without compromising pan-cardiac CTNT expression (*Figure 2G–I*). Similar observations were made upon enforced long-term (3 wk) overexpression of *ISL1* (*Figure 2—figure supplement 1E*). Collectively, these data reveal distinct stage-dependent effects of *ISL1* overexpression, in agreement with the mouse system (*Dorn et al., 2015*). Significantly, however, they establish that at early differentiation stages, ISL1 acts as a pro-differentiation factor.

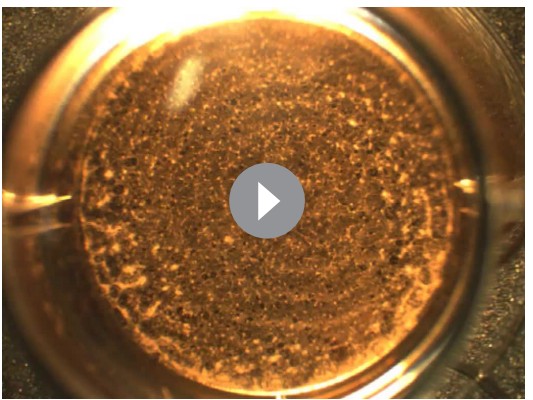 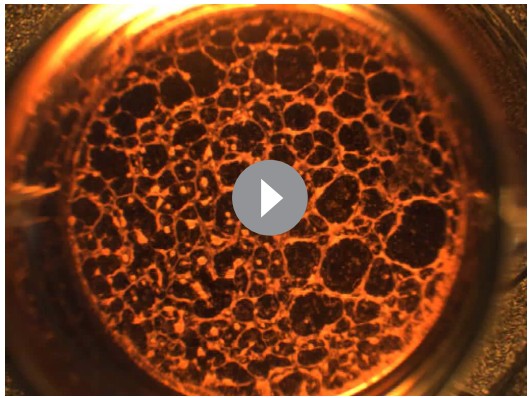

**Video 1.** Spontaneously beating WT hESC-CMs. Stereo microscopic view of WT cells at day 8 of cardiac differentiation (24-well format).
DOI: https://doi.org/10.7554/eLife.31706.004

**Video 2.** Spontaneously beating ISL1 KO hESC-CMs. Stereo microscopic view of ISL1 KO cells at day 8 of cardiac differentiation.
DOI: https://doi.org/10.7554/eLife.31706.005

## ISL1 KO phenocopies an atrial wild-type CM phenotype induced by retinoic acid

As already mentioned, ISL1 KO cells displayed a delay in beating initiation and also, at a later stage, appeared to have shorter contraction durations as compared to wild-type cells. These observations reminded us of the behavior of wild-type CMs differentiated in the presence of retinoic acid, which we coincidently studied in parallel efforts. RA treatment during CM induction is to apply a physiological signaling cue for promoting CM subtype specification into atrial cells, as opposed to ventricular CMs usually obtained by default (*Devalla et al., 2015*; *Ma et al., 2011*). Interestingly, a thorough optimization of atrial fate induction by RA in WT cells revealed that the optimal time window of exposure overlapped with that of endogenous *ISL1* induction in our protocol. Hence, a 2-day treatment with RA (0.5 µM) on days 3 and 4 was sufficient to induce robust atrial specification while suppressing a ventricular fate (*Figure 3—figure supplement 1A–D*). Concomitantly, however, RA administration caused a differentiation delay similar to ISL1 KO cells differentiated with the standard protocol (*Figure 3A and B*). We therefore hypothesized that the ISL1 KO - instead of compromising CM formation in general - may cause a switch in CM subtype identity, from ventricular to atrial-like, similar to the effects of RA stimulation.

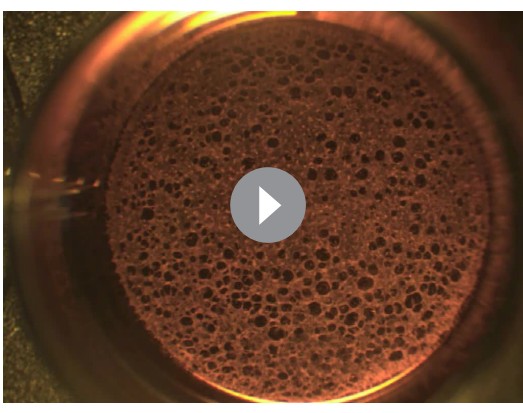

**Video 3.** Spontaneously beating ISL1 KO hESC-CMs rescued by transgenic *ISL1* induction at days 3 and 4. Stereo microscopic view of ISL1$^{KO/I.TET-ON}$ cells treated with DOX$_{d3,4}$ (day 8 of cardiac differentiation).
DOI: https://doi.org/10.7554/eLife.31706.006

A microarray analysis of later-stage ± DOX$_{d3,4}$-treated ISL1$^{KO/I.TET-ON}$ cardiomyocytes, ± RA$_{d3,4}$-treated CMs, as well as human ventricular and atrial heart tissue served to address this idea. A panel of accepted and newly identified markers such as atrial *ANP*, *DHRS9*, *KCNA5*, *SLN* (encoding sarcolipin) and the ventricular-specific *MLC2v* (*Josowitz et al., 2014*; *Piccini et al., 2015*) could well discriminate between the distinct human heart samples. Strikingly, these were also differentially expressed between DOX-treated ISL1$^{KO/I.TET-ON}$ and WT CMs on the one hand, and *ISL1*-deficient and RA-treated WT CMs on the other (*Figure 3C*). These data were confirmed in independent experiments, at RNA and protein levels: RA-treated WT CMs consistently showed a pronounced expression of atrial-specific markers and a strong decline in ventricular-specific gene expression - and this was phenocopied by the ISL1 knockout, with only some variation in the

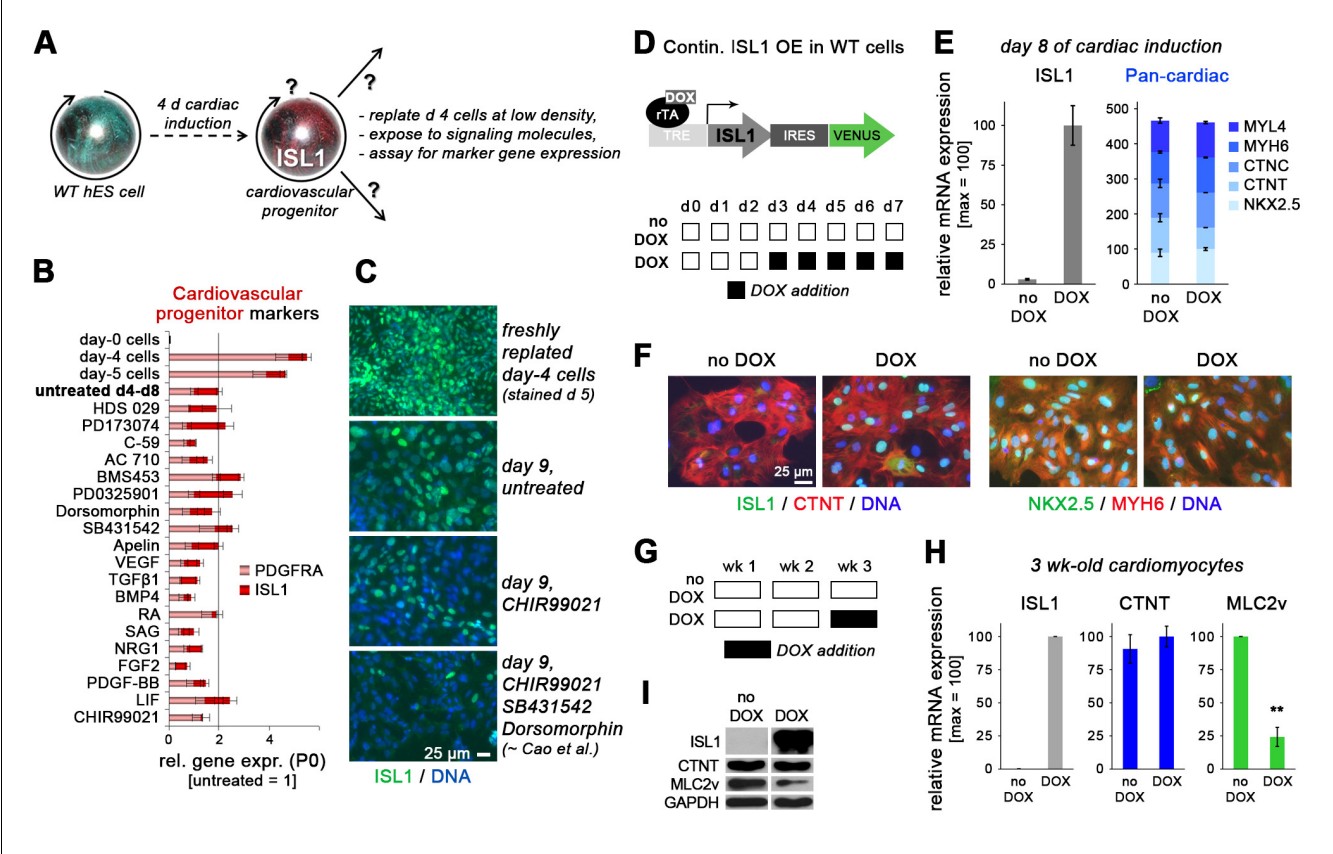

**Figure 2.** ISL1 does not stabilize the cardiovascular progenitor cell state. (**A**) Assay design interrogating self-renewal and multipotent properties of ISL1-expressing cells which emerge by day 4/5 of directed cardiac differentiation. (**B**) RT-qPCR results from screening the indicated signaling molecules for sustaining cardiac progenitor-specific gene expression within the first passage using WT cells (n = 3–4 per sample type). (**C**) ISL1 immunostaining of samples treated in the indicated ways. (**D**) Schematic of inducible expression vector used to generate a WT[I.TET-ON] hESC line (top) and doxycycline treatment protocol used here to induce a continuous overexpression of *ISL1* on ISL1[WT] background (bottom). (**E**) RT-qPCR analysis of *ISL1* (left) and a set of 5 pan-cardiac genes (right) at day 8 of directed cardiac differentiation without or with continuous DOX treatment as shown in panel D (n = 2). (**F**) Immunostaining of ISL1 and early cardiomyocyte markers at day 8 of directed cardiac differentiation. Note that endogenous ISL1 has already declined in untreated cells, whereas transgenic ISL1 does not interfere with CM formation. (**G**) Schematic of DOX treatment protocol used here to induce *ISL1* overexpression in maturing CMs. (**H**) RT-qPCR analysis in 3-week-old CMs following late DOX treatment of the indicated samples (n = 3). (**I**) Immunoblot confirming RT-qPCR analysis (**H**) at the protein level.

DOI: https://doi.org/10.7554/eLife.31706.007

The following figure supplement is available for figure 2:

**Figure supplement 1.** Multipotency of transient hESC-derived ISL1[+] cells and effects of continuous overexpression.

DOI: https://doi.org/10.7554/eLife.31706.008

degree of the phenotypes from experiment to experiment (*Figure 3D–F*). These data establish a novel and crucial role of ISL1 at the cardiac precursor state, namely, to promote ventricular CM subtype specification and counteract an atrial fate, thereby competing with RA signaling for prospective CM identity (*Figure 3G*).

Intriguingly, ISL1 depletion alone was apparently sufficient to promote an atrial-like expression pattern - without need for exogenous RA stimulation during cardiac induction. We next sought to investigate whether this change in gene expression would also translate into distinct functional properties. In addition, we wondered if RA stimulation and ISL1 depletion would synergize to yield an enhanced atrial phenotype. To this end, we compared untreated WT CMs, ISL1 KO CMs, RA$_{d3,4}$-treated WT CMs, and RA$_{d3,4}$-treated ISL1 KO CMs to one another. The latter type indeed showed enhanced expression levels of the atrial regulator *NR2F1* and of *KCNA5* which encodes a key atrial-specific potassium ion channel (*Figure 3—figure supplement 1E* and *Figure 3H*, respectively). KCNA5 accounts for the shortening of atrial CM-specific action potentials as compared to ventricular

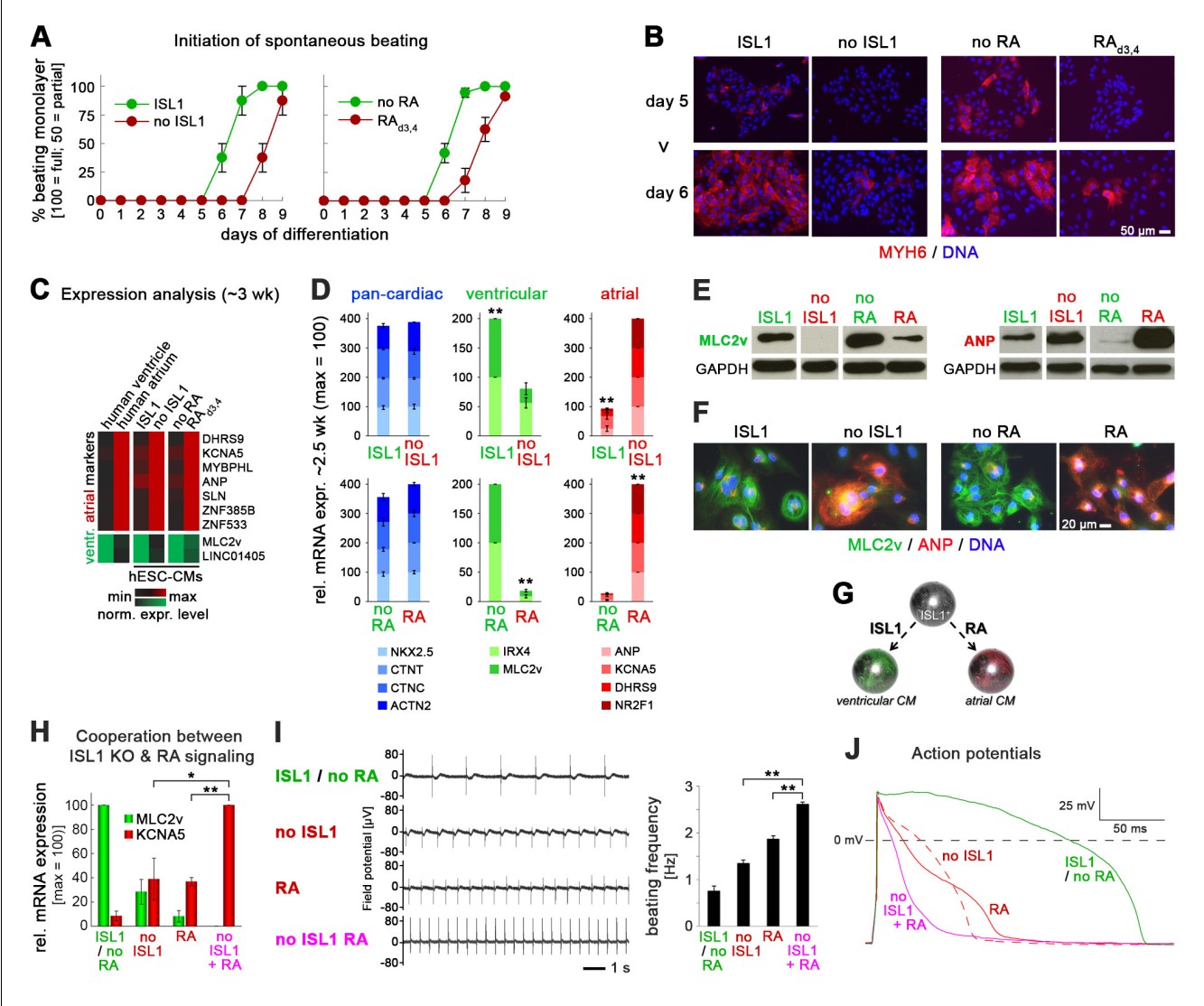

**Figure 3.** ISL1 KO phenocopies an atrial wild-type CM phenotype induced by retinoic acid. (A) Upon directed cardiac induction, ISL1 KO as well as RA-treated wild-type hESCs display delayed terminal CM differentiation reflected by a later initiation of spontaneous beating (semiquantitiative analysis, n = 3–14 per sample). 'ISL1' and 'no RA' denote different batches of WT HuES6 cells. (B) Immunofluorescence analysis of the early CM marker myosin heavy chain 6 upon directed cardiac differentiation of the indicated samples/cell lines. 'ISL1' cells are d 3/4 transgene-induced ISL1 KO hESCs. (C) Expression pattern of atrial and ventricular-enriched genes in the indicated in vivo and hESC-derived samples. Primary human heart samples served as specificity controls (microarray data). (D) Confirmation of ventricular and atrial-specific gene expression profiles by RT-qPCR in independent sets of experiments (n = 4–7 per sample type). (E) Immunoblots 3 wk after the initiation of cardiac differentiation for ventricular-specific myosin light chain and atrial natriuretic peptide. (F) Confirmation of cardiac subtype-specific phenotypes by immunostaining (~3 wk time point). (G) Model summarizing the opposing roles of ISL1 and RA signaling in cardiac subtype specification. (H) Enhanced atrial and further reduced ventricular gene expression in RA-treated ISL1 KO CMs as compared to RA-treated WT and untreated ISL1 KO cells (RT-qPCR data at ~2.5 wk, n = 3). (I) Spontaneous beating analysis of the indicated hESC-CM samples on multielectrode arrays. Left: Representative traces. Right: Beating rate quantification ($n_{tech.}$ = 3). Results were reproducible in independent experiments. (J) Representative action potential traces from patch clamp analyses of the indicated types of hESC-CMs. Note the additional action potential shortening upon combining ISL1 depletion with RA treatment. See *Supplementary file 2* for averaged data from independent samples. In case of using $ISL1^{KO/I.TET-ON}$ cells, all $ISL1^+$ data in this figure are based on a day 3–4 treatment with DOX.
DOI: https://doi.org/10.7554/eLife.31706.009

The following figure supplement is available for figure 3:

**Figure supplement 1.** Atrial specification promoted by RA stimulation or ISL1 knockout.
DOI: https://doi.org/10.7554/eLife.31706.010

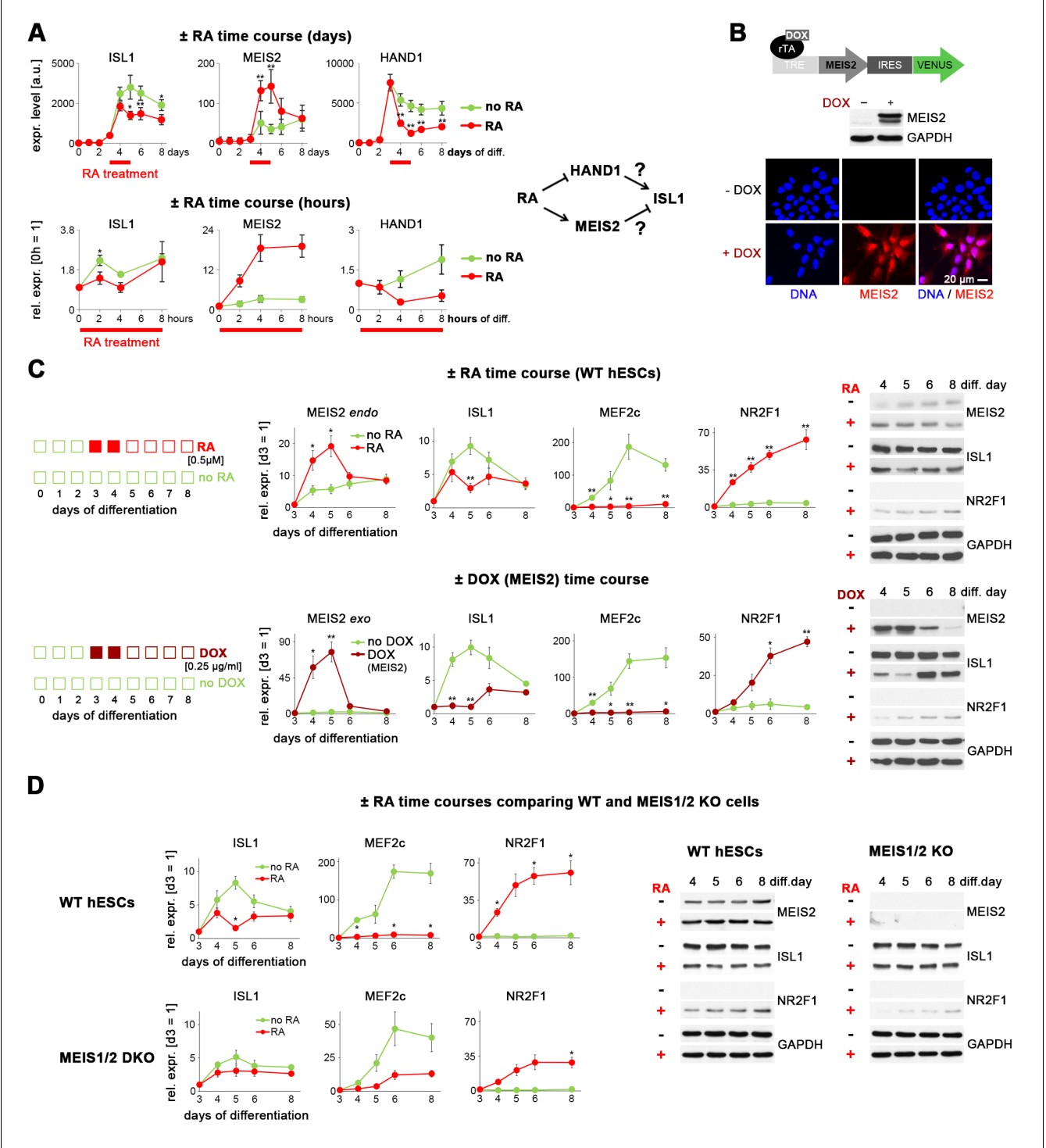

**Figure 4.** Retinoic acid signaling antagonizes *ISL1* by inducing *MEIS2*. (**A**) Top left: Microarray-based time course gene expression analysis of untreated and RA-treated (0.5 μM on d 3–4) WT hESCs subjected to cardiac induction conditions (from *Supplementary file 1*). Bottom left: RT-qPCR analysis over several hours on day 3 of cardiac induction (n = 2–3). Filtering criteria: >3 fold expression difference at day 5 between ± RA and confirmed short-term effect within 4 hr of RA treatment. Right: Deduced working hypotheses on indirect suppression of *ISL1* by RA. (**B**) Top: Schematic of PiggyBac vector for DOX-inducible *MEIS2* expression (top). Middle: Immunoblot on day 5 of cardiac differentiation confirming *MEIS2* induction in clonal WT[M.TET-ON] hESCs following DOX addition (d 3–4). Bottom: Immunostaining confirming predominantly nuclear MEIS2 (isoform D) localization. (**C**) ± RA and ± DOX (MEIS2) differentiation time courses using WT[M.TET-ON] hESCs. Left: RT-qPCR analysis (n = 4–6 each). Right: Western blot analysis. (**D**) Comparative analysis of

*Figure 4 continued on next page*

*Figure 4 continued*

HuES6 WT and MEIS1/2 KO hESC differentiation without or with RA addition (d 3–4). Left: RT-qPCR analysis (n = 3 each). Right: Corresponding immunoblot analysis.

DOI: https://doi.org/10.7554/eLife.31706.011

The following figure supplement is available for figure 4:

**Figure supplement 1.** Induced HAND1 and MEIS2 overexpression experiments and MEIS1/2 double-knockout.

DOI: https://doi.org/10.7554/eLife.31706.012

ones, which also translates into faster spontaneous beating of atrial-like hiPSC-CMs (*Marczenke et al., 2017b*). Likewise, ISL1 KO and $RA_{d3,4}$-treated hESC-CMs showed increased beating frequencies as compared to ventricular-like controls. Interestingly, $RA_{d3,4}$-treated ISL1 KO CMs contracted at even faster rates (*Figure 3I*). Furthermore, using single-cell patch clamp analysis, ISL1 KO CMs displayed atrial-type action potentials, similar to RA-CMs. In line with the increased *KCNA5* expression levels, however, action potential durations in $RA_{d3,4}$-treated ISL1 KO CMs were even further reduced (*Figure 3J* and *Supplementary file 2A*). Finally, to reveal a clear-cut functional feature, we blocked KCNA5 channels using a pharmacological inhibitor, 4-aminopyridine (*Devalla et al., 2015*). As predicted, this treatment had no effect on ventricular-like WT hESC-CMs but it significantly prolonged action potentials in the other three groups, particularly in $RA_{d3,4}$-treated ISL1 KO cells (*Figure 3—figure supplement 1F* and *Supplementary file 2B*). These results show that ISL1 KO CMs functionally resemble atrial cardiomyocytes and that this phenotype may be further enhanced by transient RA stimulation, in line with the competition model of *Figure 3G*.

## Retinoic acid signaling antagonizes *ISL1* by inducing *MEIS2*

Further, we sought to identify the gene regulatory basis of this antagonism between RA signaling and ISL1. RA signaling is known to restrict the ISL1-expressing domain at the SHF stage in vivo (*Ryckebusch et al., 2008*; *Sirbu et al., 2008*). To investigate a potentially similar effect in the hESC system, we recorded gene expression time courses of differentiating $RA_{d3,4}$-treated and untreated WT hESCs (*Supplementary file 1*). *ISL1* indeed became partially repressed by RA but it did so with a delay of about 1 day, suggesting an indirect mechanism (*Figure 4A*, left). Unbiased filtering of the array data set as well as independent short-term RA stimulation experiments revealed two genes that could potentially mediate the ISL1-suppressing effect, the transcription factor-encoding *MEIS2* and *HAND1*: *MEIS2* became immediately upregulated by RA and *HAND1* became immediately repressed (*Figure 4A*). These kinetics suggested two working hypotheses depicted in the right part of *Figure 4A*. Regardless of their specific functions in cardiogenesis reported to date, we hence decided to test both these ideas using newly generated DOX-inducible expression cell lines (*Figure 4B* and *Figure 4—figure supplement 1A*). Enforced *HAND1* induction during cardiac differentiation did not lead to an upregulation of *ISL1*, which lead us to drop the first hypothesis (*Figure 4—figure supplement 1A*).

By contrast, *MEIS2* activation, as restricted to the optimal RA treatment window at days 3 and 4 of directed cardiac differentiation, significantly - and reversibly - suppressed *ISL1*, similar to RA treatment itself (*Figure 4C*). The downstream effects on the predicted human ISL1 target gene *MEF2C* were even more severe. Importantly, *MEIS2* induction also caused an upregulation of *NR2F1* which is a key driver of atrial fate induction, albeit with some delay compared to RA stimulation (*Figure 4C*; *Devalla et al., 2015*). Moreover, all these effects were RA signaling and *MEIS2* gene dose-dependent (*Figure 4—figure supplement 1B*). Hence, MEIS2 appears to be an important player in atrial CM specification, acting downstream of RA signaling to suppress *ISL1* and promote the induction of the atrial specifier gene *NR2F1*.

To further strengthen this conclusion, we next sought to perform the reverse experiment, after disrupting *MEIS2* together with *MEIS1* that might otherwise compensate for the loss of its family member (*Figure 4—figure supplement 1C*). Clonal MEIS1/2 double-knockout (DKO) hESCs were subjected to directed cardiac induction in comparison to WT cells. Terminally differentiated MEIS1/2 DKO cells failed to show spontaneous beating, yet they stained positive for prominent pan-CM markers (*Figure 4—figure supplement 1D*). We reasoned that in light of the mildness of this DKO phenotype, the model would still allow for investigating the early regulatory relationships of interest

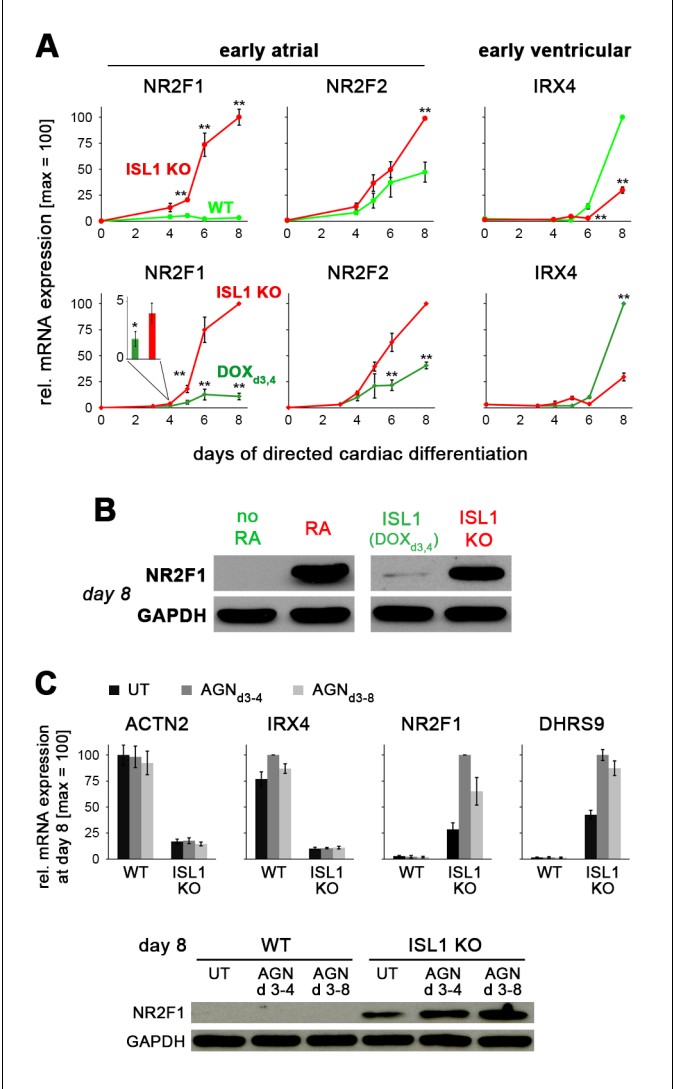

**Figure 5.** ISL1 suppresses the early atrial specifier *NR2F1*. (**A**) Time course gene expression analysis of early atrial and ventricular genes by RT-qPCR (n = 2–10 per data point). Comparison between WT and ISL1 KO cells (top), and ISL1 KO versus pulsed ISL1 rescue (bottom, with ISL1$^{KO/I.TET-ON}$ cells). Note the immediate-early effect on *NR2F1*, which is not seen in case of *NR2F2* or *IRX4*. (**B**) Immunoblot for NR2F1 at an early CM stage. (**C**) RA receptor antagonist AGN 193109 (100 nM) does not rescue the phenotype of ISL1 KO cells, as shown by RT-qPCR (top, n = 4) and immunoblot (bottom) in the indicated conditions.

DOI: https://doi.org/10.7554/eLife.31706.013

The following figure supplement is available for figure 5:

**Figure supplement 1.** ISL1 represses the atrial program in favor of ventricular specification.

DOI: https://doi.org/10.7554/eLife.31706.014

upon precursor cell formation. In WT controls, RA administration at days 3 and 4 partially suppressed ISL1 and promoted *NR2F1* induction, as seen before (*Figure 4D*). In comparison, MEIS1/2 DKO cells failed to markedly suppress *ISL1* as well as its target gene *MEF2C* in the presence of RA, and the RA-mediated induction of *NR2F1* was strongly compromised (*Figure 4D*). These data establish MEIS2 as a mediator of *ISL1* suppression by RA signaling, thereby favoring atrial specification.

## ISL1 suppresses the early atrial specifier *NR2F1*

Conversely, we asked how ISL1 may antagonize atrial induction promoted by RA signaling. In addition to *NR2F1*, the family member *NR2F2* as well has been shown to be an important atrial driver

Developmental Biology and Stem Cells

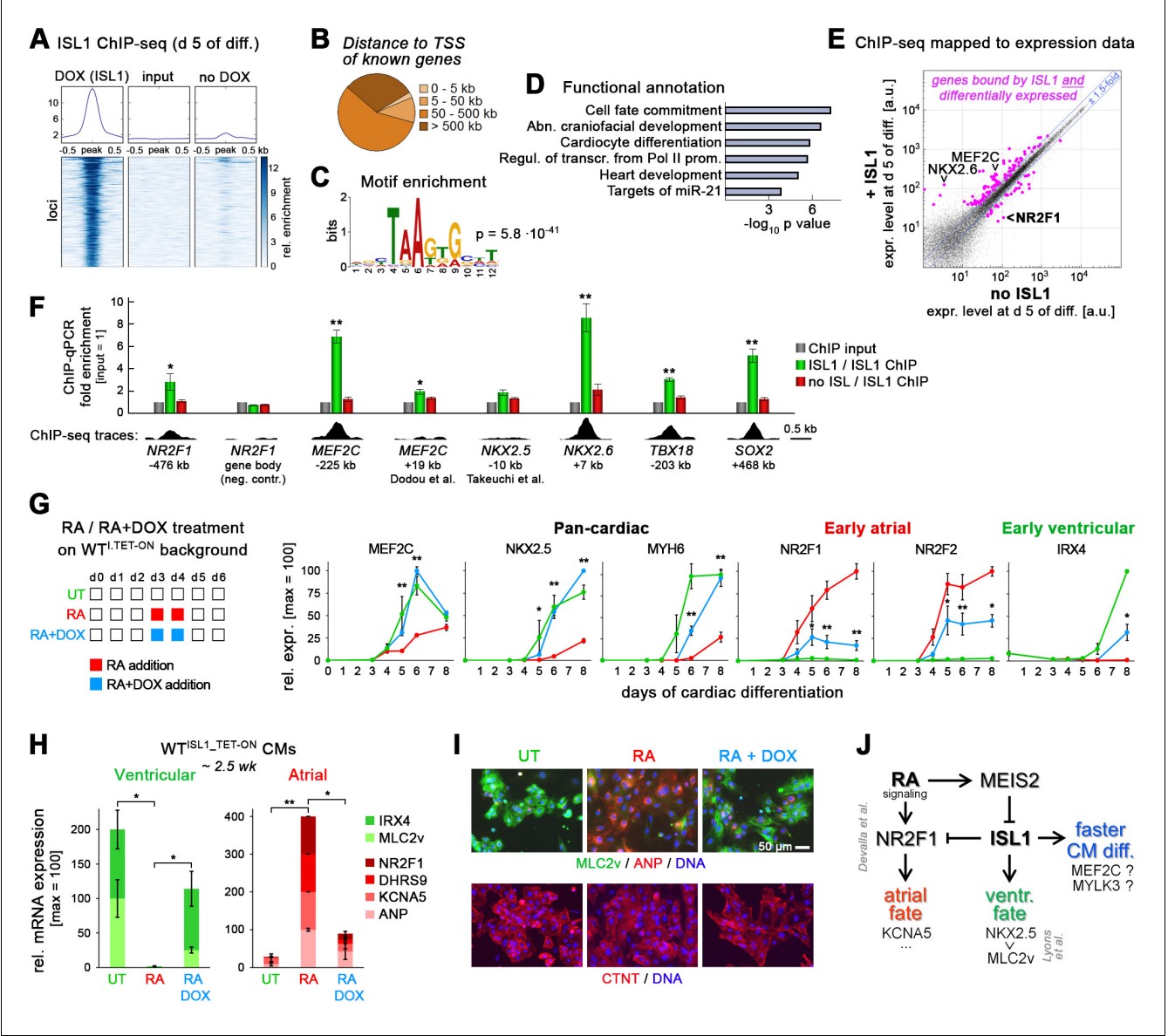

**Figure 6.** ISL1 functionally antagonizes atrial specification driven by RA signaling. (A) ISL1 ChIP-sequencing in differentiating ISL1[KO/I.TET-ON] cells. Summary plot of normalized scores for ISL1 peak regions (top) and intensity-sorted heat map for detected peaks called in DOX-treated samples versus ChIP input DNA (also see *Supplementary file 3*). No-DOX cells served as an additional specificity control. Minor signals in this sample likely result from leaky transgene expression. (B) ISL1-bound peak distribution relative to transcription start sites of known genes. (C) Single enriched motif identified using DNA sequences underlying ISL1-bound peak regions. (D) Functional annotation of gene set associated with ISL1 peaks (also see *Supplementary file 3*). (E) Highlighting of ISL1-bound genes in a scatter plot revealing *ISL1*-induced differential gene expression at day 5 of differentiation (line ISL1[KO/I.TET-ON], ± DOX treated at d 3–4). Expression ratio cutoff: 1.5-fold. Also see color-coded data in *Supplementary file 3*. (F) ChIP-qPCR analysis in differentiating ISL1[KO/I.TET-ON] hESCs of newly identified ISL1-bound sites as well as of regions homologous to published mouse ISL1 enhancers (n = 4–5). See *Supplementary file 4* for amplicons used. Bottom: Corresponding ISL1 ChIP-seq pileups. (G) Left: Design of RA/ISL1 competition experiment using WT[I.TET-ON] cells. Right: RT-qPCR analysis (n = 4). (H) CM fate analysis of RA/ISL1 competition experiment at 2.5 wk (RT-qPCR data, n = 4). (I) CM subtype analysis of the indicated samples 3 wk after differentiation start. (J) Elucidated regulatory module controlling cardiac subtype specification and CM formation speed in differentiating hESCs. See text for discussion. The RA-NR2F1-KCNA5 axis has previously been revealed by *Devalla et al. (2015)*.

DOI: https://doi.org/10.7554/eLife.31706.015

The following figure supplement is available for figure 6:

**Figure supplement 1.** ISL1-controlled gene expression in differentiating hESCs.

DOI: https://doi.org/10.7554/eLife.31706.016

downstream of RA (*Devalla et al., 2015*). We hence performed time course expression analyses comparing ISL1⁺ and ISL1-deficient hESCs undergoing cardiac induction, while paying particular attention to these two genes. Other early or later atrial and ventricular markers did not consistently show an immediate response to *ISL1* activation by DOX in differentiating ISL1^{KO/I.TET-ON} cells (*Figure 5A*, right, and *Figure 5—figure supplement 1A*). This was also true for *NR2F2* which showed a significant but overall delayed suppression response to ISL1 (*Figure 5A*, middle panel). By contrast, *NR2F1* became rapidly induced in ISL1 KO cells, remained essentially unexpressed in WT cells, and was antagonized by DOX-mediated ISL1 induction as early as at day 4 (*Figure 5A*, left). Accordingly, pulsed *ISL1* induction by DOX addition on days 3 and 4 of differentiation preserved low levels of NR2F1 protein in early CMs (*Figure 5B*). These data show that ISL1 serves to antagonize the key atrial regulator gene *NR2F1*.

Further, we sought to test whether endogenous RA signaling is involved in promoting *NR2F1* induction in an ISL1-deficient setting. We hence employed AGN 193109 (AGN), a pan RA receptor antagonist (*Agarwal et al., 1996*). AGN effectively antagonized RA-driven atrial induction in a dose-dependent manner (*Figure 5—figure supplement 1B*). Based on these validation data, differentiating ISL1 KO hESCs were exposed to the RA inhibitor. 100 nM AGN did not compromise (but rather enhanced) *NR2F1* induction caused by ISL1 deficiency (*Figure 5C*). Moreover, transient *ISL1* overexpression in differentiating ISL1^{KO/I.TET-ON} cells did not lead to the downregulation of *MEIS2*, which, as shown above, acts downstream of RA signaling to downregulate *ISL1* and upregulate *NR2F1* (*Figure 5—figure supplement 1C*). These experiments suggest that ISL1 controls a RA-independent axis in suppressing *NR2F1*.

## ISL1 functionally antagonizes atrial specification driven by RA signaling

The above data uncover an important function of ISL1 in repressing atrial specification, and also in accelerating CM formation in favor of a ventricular identity. To understand whether these effects are likely to be based on direct gene regulation, we performed ISL1 ChIP-sequencing based on day 5 differentiating samples. In employing ISL^{KO/I.TET-ON} cells, no-DOX samples could serve as a near-perfect specificity control. Surprisingly, the number of unequivocally ISL1-bound sites was rather limited (~200 loci), while technical reasons cannot be ruled out to account for this result (*Figure 6A* and *Supplementary file 3*). Most of these hits were rather distant to transcription start sites of known genes (*Figure 6B*). A sequence analysis of the bound regions, however, identified a short binding motif resembling those obtained in the mouse neural context (*Figure 6C*; *Mazzoni et al., 2013*). Moreover, functional annotation of the nearest genes revealed biologically meaningful terms like 'cardiocyte differentiation' and 'heart development', and interestingly also that ISL1 binding was enriched near genes with regulatory functions (*Figure 6D*).

To assess which binding events may translate into actual changes in cardiac-associated gene expression, the target gene set of *Supplementary file 3* was mapped to the day 5 data of *Supplementary file 1*, that is, to ISL1-dependent differential gene expression at the cardiac precursor state. This functionally regulated subset also included *MEF2C*, a crucial cardiac regulator downstream of ISL1 in mouse (*Wang et al., 2016*). Although the fold enrichments were rather low, independent ChIP-qPCR analysis clearly confirmed several binding events associated with differential gene expression at day 5, which also included *NR2F1* (*Figure 6F*). We also noticed distal ISL1 binding near the anti-mesodermal *SOX2* locus (*Rao et al., 2016*), implying that ISL1 may help to sustain its repression during cardiac induction. Indeed, ISL1 KO cells tended to moderately regain *SOX2* expression and *ISL1* could antagonize this gene when activated in undifferentiated hESCs (*Figure 6—figure supplement 1A and B*). By contrast, following up on the putative ISL1-controlled repressor element upstream of *NR2F1* - by excising it from the genome of ISL1^{KO/I.TET-ON} cells - did not bear obvious functional implications (*Figure 6—figure supplement 1C*). Overall, these data suggest that ISL1 acts by regulating a limited but crucial set of downstream regulatory genes. *NKX2.5* was not picked up by our ChIP-seq analysis although transcriptionally, it fulfilled all criteria of being direct human ISL1 target as well (*Figure 1—figure supplement 1B and D*).

Finally, we sought to assess whether or not the repression of *ISL1* by retinoic acid signaling revealed above presents a functional requirement for atrial induction. To this end, we designed a competition experiment between RA and ISL1 in WT^{I.TET-ON} cells (*Figure 6G*, left). Stage-specific *ISL1* overexpression during pro-atrial RA supplementation promoted WT-like CM differentiation kinetics and strongly interfered with atrial gene induction - and with the upregulation of *NR2F1* in

particular (*Figure 6G*, right, and *Figure 6—figure supplement 1D*). Analysis of the resulting CMs revealed a ventricular rather than an atrial subtype identity of the RA+DOX cells, indicating that ISL1 dominates the experimental outcome in this scenario (*Figure 6H and I*).

## Discussion

This study uncovers previously unrecognized functions of ISL1 in cardiac differentiation and, particularly, it assigns a central role to this gene in the fundamental context of cardiac subtype specification. Notably, our results are exclusively based on the human ES system and thus, it is presently uncertain to which extent they may also apply to the in vivo context.

### Differentiation-promoting function of ISL1

Based on its mouse knockout phenotype and its pronounced expression in the expanding SHF, ISL1 has thus far been associated with the proliferating, that is self-renewing, state of cardiac precursor cells (*Cai et al., 2003*; *Cohen et al., 2007*; *Laugwitz et al., 2008*; *Zhang et al., 2016*). Indeed, isolated ISL1$^+$ cells have been shown to be amenable to (limited) expansion in vitro (*Bu et al., 2009*; *Cao et al., 2013*; *Qyang et al., 2007*). However, ISL1 merely served as a marker of the cardiac precursor cell state in these studies, which does not necessarily imply any active role in sustaining self-renewal of these populations. Hence, our finding that transient *ISL1* induction, and even its continuous enforced expression, does not block the differentiation process but actually accelerates the induction of prominent cardiomyocyte-specific genes, may seem surprising but does not per se present a conflicting result with previously published work. In support of this view, *Kwon et al. (2009)* proposed that ISL1 may actually be counterproductive for sustaining the cardiac precursor state.

Interestingly, our observations in the hESC system are also confirmed by enhanced cardiomyocyte yields obtained after constitutive *Isl1* overexpression in mouse ES cells, suggesting a universal cardiac differentiation-promoting role of this factor (*Dorn et al., 2015*; *Kwon et al., 2009*). How is this effect brought about? Our ChIP-seq analysis in conjunction with our differential time course expression data suggests that ISL1 does not immediately promote the induction of structural CM genes: Although some of these - *MYH6*, *MYL4*, and others - start becoming upregulated as early as at day 4/5 of directed cardiac induction, their induction patterns were either unaffected by depleting/reintroducing *ISL1* or we did not observe any ISL1 binding to their loci, with only few putatively minor exceptions (*MYLK3*; *Chan et al., 2008*). Rather, ISL1 appears to affect the expression of a small but crucial set of transcriptional regulators in differentiating hESCs, notably *MEF2C* and *NKX2.5* (and *NKX2.6*) - albeit likely via an alternative human-specific enhancer in case of *MEF2C* and with borderline evidence for direct binding in case of *NKX2.5* (*Dodou et al., 2004*; *Lin et al., 1997*; *Lyons et al., 1995*; *Takeuchi et al., 2005*; *Tanaka et al., 2001*). Interestingly, several studies have revealed a negative feedback mechanism by mouse NKX2.5 on the cardiac progenitor state and on *Isl1* in particular (*Dorn et al., 2015*; *Prall et al., 2007*; *Watanabe et al., 2012*), which is also in line with the rapid decline of *ISL1* in early NKX2.5$^+$ hESC-CMs (*Zhang et al., 2015*). This mechanism might therefore imply that cardiac *ISL1* expression is transient by nature, that is, inevitably fated to promote a successive conversion into an early cardiomyocyte identity. Thus, attempts to stabilize the ISL1$^+$/NKX2.5$^-$ cardiac precursor cell state long-term in a cell culture setting would need to neutralize the pro-differentiation function of ISL1, which has thus far failed in our hands using candidate signaling molecules or cocktails thereof (*Cao et al., 2013*; *Cohen et al., 2007*; *Qyang et al., 2007*).

### Paradigm for second heart field development?

A compelling question is whether hESCs on their way to differentiating into cardiomyocytes actually take a FHF or a SHF-like route. At first sight, ISL1 itself could be considered as a marker to address this issue because it is a prime player in the SHF. Sensitive Cre driver lines, however, additionally suggest a transient expression of *Isl1* in the FHF (*Dorn et al., 2015*; *Prall et al., 2007*). To the best of our knowledge, the very transient expression of *Isl1* in the FHF - and likewise in the posterior SHF - is not associated with any major function (*Cai et al., 2003*; *Dodou et al., 2004*). Therefore, our observed impairment of cardiac induction following of *ISL1* depletion in hESCs would best fit to the idea that hESCs tend to differentiate via a SHF-like path - at least in our particular protocol. In general, however, there is a scarcity of markers definitely distinguishing the SHF from the FHF and hence, this hypothesis seems hard to substantiate. It is noteworthy, at least, that the expression of

some SHF-associated genes like *FGF8*, *FGF10*, or *HAND2* was indeed compromised in our ISL1 KO time course (*Supplementary file 1*). Nonetheless, it may also be that even tightly controlled hESC differentiation protocols do not strictly adhere to in vivo developmental routes. Moreover, it is conceivable that different differentiation protocols may give rise to distinct differentiation intermediates or mixtures thereof (*Lian et al., 2012*; *Yang et al., 2008*; *Zhang et al., 2015*). Additional loss- and gain-of-function studies in the hESC system - targeting *MEF2C* and others - are needed to better understand the degree of similarity between in vivo cardiogenesis and directed cardiac hESC differentiation.

## Antagonism with RA signaling and cardiac subtype-specification module

hESCs undergoing cardiac induction are responsive to RA and the temporal window of opportunity for RA-mediated atrial specification overlapped with the induction of *ISL1*. Besides inducing a key regulator of atrial fate induction, *NR2F1* (and NR2F2; *Devalla et al., 2015*), we show that RA antagonizes *ISL1* expression, which appears to be in agreement with an expansion of the Isl1$^+$ anterior SHF domain following RA signaling depletion in vivo (*Ryckebusch et al., 2008*; *Zaffran et al., 2014*). Restriction of *Isl1* by RA in mouse and/or zebrafish is thought to be mediated via a repression of FGF signaling and/or via an induction of the LIM domain protein-encoding gene *Ajuba* (*Sirbu et al., 2008*; *Witzel et al., 2012*). AJUBA (*JUB*) was only expressed at low levels in our ± RA differentiation time courses and was not positively regulated by RA. Furthermore, *FGF8* did become repressed by RA but FGF signaling tended to affect *ISL1* expression levels in a negative rather than in a positive manner, as based on the aforementioned assay. Instead, we find that in differentiating hPSCs, the slightly delayed and transcriptionally moderate repression of *ISL1* by RA is mediated by MEIS2, a known but still underinvestigated player at the cardiac precursor stage (*Dupays et al., 2015*; *Paige et al., 2012*; *Wamstad et al., 2012*). Hence, we propose that RA exerts its *ISL1*-repressing function through this indirect mechanism, at least in the human ESC system (*Figure 6J*). Interestingly, though, the regulatory link between RA signaling and *Meis2* has also been revealed in a different context, limb development, raising the possibility that it might also be operative in the developing heart (*Cunningham and Duester, 2015*).

Another key finding of this study is that ISL1 functionally serves to antagonize atrial induction, most immediately by suppressing the upregulation of *NR2F1* (*Devalla et al., 2015*), in a RA-independent fashion. Importantly, our competition experiments indicate that the enforced expression of *ISL1* may neutralize the pro-atrial effects of RA. These data therefore reveal that *ISL1* repression is a key functional requirement - not merely a correlative event - for enabling atrial induction. Furthermore, ISL1 depletion and RA signaling synergized to give rise to cells with enhanced atrial features. The antagonism between RA signaling and ISL1 converging on *NR2F1* hence appears to form the core of a regulatory module controlling cardiac subtype specification (*Figure 6J*).

Moreover, overlapping with the delay in pan-cardiac differentiation, *ISL1* disruption alone was sufficient for promoting a mild atrial-like CM phenotype at the expense of a ventricular identity. According to the model, this would primarily be mediated through derepression of *NR2F1*, while we speculate that autocrine RA signaling might serve as an additional intrinsic cue in this scenario (*Lee et al., 2017*). Interestingly, *Isl1*-deficient SHF cells in vivo fail to form right ventricular progeny, yet they show sustained competence for atrial specification (*Cai et al., 2003*). Hence, our hESC-based findings - that *ISL1* is a key pro-ventricular/anti-atrial player - appear to conform with observations made in the mouse embryo. As a seeming discrepancy with our data and interpretation, though, *Dorn et al. (2015)* observed a suppression of ventricular fate following *Isl1* overexpression in differentiating mouse ES cells. However, this discordance may be explained by the fact that these authors used long-term *Isl1* overexpression, as opposed to our mimicking of the endogenous *ISL1* expression pattern using pulsed - not constitutive - transgene induction: Indeed, ISL1 induction in already formed CMs or long-term overexpression confusingly suppressed ventricular-specific markers, similar to observations made by Dorn and colleagues using mouse cells. We think, though, that this treatment disregards the transient nature of *ISL1* expression upon cardiac induction and therefore, this late anti-ventricular action should be considered non-physiological.

This study exemplifies the utility of combining genetic manipulation of hESCs with their controlled differentiation, as applied to deciphering the gene regulatory network underlying cardiac subtype specification. It will be interesting in future to study a number of remaining questions, such as the exact mode of *NR2F1* suppression by ISL1 or the later-stage repression of ventricular-specific gene

induction (*MLC2v*, *IRX4*) by the atrial program. Besides illuminating a fascinating developmental paradigm, these efforts will also help to further improve cardiac subtype-specific differentiation protocols for applied biomedical purposes (*Marczenke et al., 2017b*).

## Materials and methods

### Cell culture and differentiation

HuES6 hESCs (*Cowan et al., 2004*) served as starting material for genetic manipulation and experimentation. RA-driven atrial induction was independently reproduced using F1 hiPSCs (*Marczenke et al., 2017a*). hESCs were routinely maintained in defined FTDA medium on 1:75 diluted Matrigel HC (Corning # 354263; *Frank et al., 2012*). FTDA was composed of DMEM/F12, 1% (v/v) PenStrep/Glutamine, 1% (v/v) defined lipids (Thermo # 21331020, # 10378016, and # 11905031, respectively), 0.1% (v/v) ITS (Corning # CB-40350), 0.1% (w/v) human serum albumin, 10 ng/ml FGF2, 0.2 ng/ml TGFβ1, 50 nM Dorsomorphin, and 5 ng/ml Activin A (also see *Supplementary file 4*). Some genetically modified lines were additionally supplemented with 2 nM C-59 to fully suppress spontaneous differentiation. Fully confluent hESC cultures were routinely passaged using single-cell dissociation with Accutase and reseeded into new 6-well plates at 400,000 cells per well in the presence of 10 µM Y-27632. Cells were split every 3–4 days and kept in culture for a maximum of 30 passages. hESCs were previously confirmed to maintain a normal karyotype during this time frame (in *Zhang et al., 2015*) and tested negative for mycoplasma.

Cardiac induction was carried out in a monolayer format under defined serum and serum albumin-free conditions (*Zhang et al., 2015*). Fully confluent day 3 or day 4 hESC cultures - which then appeared as a tight epithelial layer - were harvested using Accutase, centrifuged at 200 g, resuspended in day-0 differentiation medium, and seeded out at 500,000 cells per well of a Matrigel-coated 24-well plate, in 2 ml of medium. Day-0 differentiation medium consisted of Knockout DMEM (Thermo), ITS, PenStrep/Glutamine, 10 µM Y-27632, 20 ng/ml FGF2 (or 10 ng/ml FGF2/5 ng/ml Activin A), 0.5–1 ng/ml BMP4, and 1 µM CHIR99021 (also see *Supplementary file 4*). Differentiation medium was exchanged on a daily basis. From day 1 onwards, basal differentiation medium consisted of KO-DMEM, 1% (v/v) TS (transferrin/selenium), 250 µM 2-phospho-ascorbate, and PenStrep/Glutamine. TS stock was prepared in advance by dissolving 55 mg transferrin (Sigma-Aldrich # T8158) in 100 ml PBS containing 0.067 mg sodium selenite. WNT inhibitor C-59 (0.2 µM) was added on days 2 and 3 to the differentiating cultures for promoting cardiac specification. In transgenic cell lines, *ISL1* or *MEIS2* were induced using a minimum dose of doxycycline leading to near-homogeneous expression in the cell populations (0.2–1 µg/ml), and typically in such a way that transgene expression matched the temporal induction patterns of the corresponding endogenous factors in WT cells. Unless stated otherwise, the endogenous *ISL1* expression pattern was mimicked by pulsed DOX administration on days 3 and 4 using ISL1$^{KO/I.TET-ON}$ cells. For atrial specification using retinoic acid, RA was typically supplemented at 0.5 µM on days 3 and 4 of differentiation, in line with the optimization data of *Figure 3—figure supplement 1A–D*. Aliquots of RA stock solutions dissolved in DMSO were stored at −80°C and discarded after one-time use.

Cardiac samples were either harvested for analysis directly from primary differentiation plates or replated using TrypLE Select (Thermo) onto Matrigel-coated dishes for longer term culture and maturation. CM splitting medium consisted of RMPI 1640 (Thermo), ITS, 0.1% HSA, defined lipids, 250 µM 2-phospho-ascorbate, 0.004% (v/v) thioglycerol, PenStrep/Glutamine, and 10 µM Y-27632. CM maintenance medium was composed of KO-DMEM supplemented with ITS, HSA, defined lipids, phospho-ascorbate, thioglycerol, and PenStrep/Glutamine. Culture medium of CM maintenance plates was exchanged every few days. Cells were harvested for analysis after a 1–2.5 week maturation period following primary cardiac differentiation.

In some experiments, cells were exposed to various stimuli before reaching a definite cardiac fate. To this end, undifferentiated cells or differentiation intermediates were replated onto Matrigel-coated dishes at an assay-compatible density and treated as indicated using defined ITS/HSA medium (DMEM/DF12 plus ITS, HSA, defined lipids, and PenStrep/Glutamine).

## Genetic manipulation of hESCs

For gene disruption using the CRISPR/Cas9n system, two pairs of nickase CRISPRs were designed to encompass genomic regions of interest ('4 n' approach; *Ran et al., 2013*). CRISPR vectors were generated following a protocol associated with Addgene plasmid # 42335 (*Cong et al., 2013*). Guide RNA-specific targeting sequences are given in *Supplementary file 4*. Sense and antisense oligonucleotides composed of these sequences as well as flanking overhangs for cloning were phosphorylated in vitro and annealed in pairs, to yield double-stranded DNA fragments. These were cloned into a modified version of the pX335 vector carrying a GFP-2A-puromycin selection cassette. For deleting one genomic region, hESCs were transfected with the four corresponding CRISPR vectors using Fugene HD (200,000 cells / 2 µg of vector mix / 6 µl Fugene HD, in FTDA medium). For simultaneously disrupting two genes, hESCs were accordingly transfected with a cocktail of eight plasmids. One day later, transfected cells were enriched using transient puromycin treatment for 1 day (0.5 µg/ml). Two days later, semiconfluent cultures were replated at clonal density and fed for about 2 weeks. Half the cells from single emerging colonies were used for crude DNA isolation. Conventional PCR spanning the putative genomic deletion region was used to screen these samples for positive cell clones. The remaining half-colonies from positive clones identified this way were used for replating and expansion. PCR fragments spanning the deletion mutations were cloned and sequenced. Homozygous knock-outs of genes of interest were further confirmed at the RNA and protein levels. 2–3 KO clones per targeted locus were initially validated to yield comparable experimental outcomes, to then focus efforts on one selected clone each.

Clonal DOX-inducible overexpression lines were generated using PiggyBac transposition. ORFs of genes of interest were TOPO-cloned from cDNA of differentiating hESCs (*Supplementary file 4*). Following validation by sequencing, these fragments were subcloned into the previously used inducible expression vector KA0717 (*Rao et al., 2016*). A given construct was then co-transfected with transactivator and transposase-encoding vectors into hESCs using Fugene HD (DNA mass ratio: 10:1:3, respectively). Stable transgene-positive cells were selected using 50 µg/ml G418 and replated at clonal dilution. Emerging colonies were test-induced for 1 day using doxycycline. Single clones showing homogeneous transgene expression on the basis of an IRES-VENUS cassette were picked, expanded, and validated to still be capable of differentiation into the cardiac lineage. Transgene expression levels in individual cell lines were assessed using RT-qPCR and/or immunostaining. DOX concentrations used in experiments were based on such titration experiments and kept at a minimum level just enabling near-homogeneous transgene induction within the cell population.

## RT-qPCR and genome-wide expression analysis

Following RNA isolation with on-column gDNA digestion (Machery Nagel), reverse transcription was performed using M-MLV reverse transcriptase (Affymetrix # 78306) with oligo-dT$_{15}$ priming at 42°C. RT-qPCR was carried out using validated primers given in *Supplementary file 4* and BioRad iTaq Universal SYBR Green Supermix (# 172–5124) on an ABI 7500 cycler. *RPL37A* served as a housekeeping control standard. qPCR reactions contained 10 µl of iTaq mix, 3 µl primer mix (containing 2.5 µM of each oligonucleotide), and 7 µl pre-diluted cDNA. The ΔΔCt method was used to calculate relative transcript abundance against an indicated reference. Alternatively, results were expressed as percentage of *RPL37A* expression ($100 * 2^{-\Delta Ct}$), given a uniform primer design and careful validation with regards to amplification efficiencies. Unless otherwise stated, error bars denote standard errors between biological replicates from independent experiments. Paired/unpaired, one- or two-sided t-tests were performed as and where appropriate, based on absolute % *RPL37A* values or relative transcript abundance. * indicates a significance level of $p < 0.05$, ** indicates $p < 0.01$.

For genome-wide expression analysis, labeled cRNA was prepared from 500 ng of DNA-free RNA samples using TotalPrep linear RNA amplification kits (Thermo # AMIL1791). Microarray hybridizations on Illumina V4 human HT-12 bead arrays were carried out as recommended by the manufacturer, with 14 hr of in vitro transcription. Cy3-stained chips were scanned using HiScan SQ instrumentation. Background subtraction, cubic spline normalization, and scatter plot analysis were carried out using GenomeStudio software. Statistics were based on an implemented Illumina custom model with multiple testing corrections. Processed data were filtered in MS Excel by setting experience-based thresholds for expression changes and minimal gene expression levels. Human heart data were taken from a previous analysis (*Piccini et al., 2015*). Hierarchical clustering of genes was

performed with one minus correlation metrics and the unweighted average distance (UPGMA), also known as group average, linkage method. Violin plots were generated from filtered gene sets using a Matlab-based algorithm based on a modification of the Matlab function distribution Plot.m by Jonas Dorn, and graphics tools for calculating distributions of relative gene expression differences . For functional annotation, array probe sequences were converted into GRCh37/hg19 genome coordinates employing the Ensembl BioMart interface to then be used as input for GREAT (*McLean et al., 2010*). Statistically significant hits were subjectively filtered for biological relevance and presented based on the obtained p values. Expression levels of individual genes are presented as array intensity signals ± bead standard deviation. Full data sets comprising comparative differentiation time-courses of WT, ISL1 KO, and rescued cells have been submitted to the NCBI GEO database in a MIAME-compliant format, under accession number GSE100592.

## Immunofluorescence analysis and immunoblotting

Immunofluorescence analysis was carried out using standard procedures. Briefly, cells were fixed in culture plates using 4% (w/v) paraformaldehyde, permeabilized and blocked with 0.2% (v/v) Triton X-100 / 5% (v/v) FCS / 2% (w/v) BSA / 2% (w/v) glycine in PBS for 45 min, and incubated with primary antibodies over night at 4°C, in 0.5% BSA/PBS. Antibodies used are listed in *Supplementary file 4*. Secondary Alexa-488 or Alexa-568-conjugated antibodies and Hoechst were used for fluorescence staining of samples. Images shown are full or cropped frames taken with a 10x or 20x objective, as appropriate, mounted to a Zeiss Axiovert microscope. Cellular morphology was captured using Olympus CKX41 cell culture or Leica MZ16 stereo microscopes.

For western blotting, protein lysates were prepared for 20 min on ice using NP-40-containing RIPA buffer with protease inhibitors and benzonase. Total protein concentrations were adjusted using Bradford assays. SDS-PAGE electrophoresis using 10–80 μg of protein per sample and electroblotting were performed employing standard procedures and equipment (BioRad). Nitrocellulose membranes were blocked with 4% (w/v) milk in 0.1% (v/v) Tween 20/PBS. Most primary antibodies (*Supplementary file 4*) were diluted in 0.5% BSA/PBS T. HRP-conjugated secondary antibodies and West Pico chemiluminescent substrate (Thermo # 34087) were used for protein detection on x-ray films.

## FACS analysis

For cell cycle analysis, differentiating hESCs were dissociated into single cells using TrypLE Select (Thermo). After dissociation, cells were pelleted, washed in PBS, resuspended in 50% FCS/PBS and fixed in 70% ethanol for 24 hr. Then, cells were incubated for 3 hr with 20 μg/ml propidium iodide (Sigma-Aldrich) and 200 μg/ml DNase free RNase A in 0.1% (v/v) Triton X-100/PBS, and analyzed on a Beckman Coulter Gallios device.

## ChIP-qPCR and ChIP-Seq

For chromatin immunoprecipitation, differentiating day 5 cells were harvested using Accutase, resuspended in ITS/HSA medium (see above) and fixed for 10 min by adding freshly thawed formaldehyde solution to a final concentration of 1% (w/v). After stopping with 125 mM glycine, pellets of $5 * 10^6$ cells were stored at −80°C or processed immediately. Samples were lysed in hypotonic Tris buffer and sonicated 35 times with a 30 s on / 30 s off cycling protocol using a Diagenode Bioruptor device at high power (volume: 300 μl), to yield an average DNA fragment size of ~200 bp. Following 10-fold dilution with 50 mM Tris pH 8 / 200 mM NaCl / 5 mM EDTA / 0.5 (v/v) NP40, sheared samples were incubated with 10 μg of antibody over night at 4°C on a rotator. Antibodies used were raised against a transgenic ISL1-HA tag or against ISL1 itself (*Supplementary file 4*). Antibody/chromatin complexes were isolated using Pierce protein A/G-conjugated magnetic beads (Thermo # 26162) and washed repeatedly with high-salt (500 mM NaCl) wash buffer. ChIP samples were eluted using 65°C 1% (w/v) SDS-containing buffer, crosslink-reversed in the presence of 200 mM NaCl for 4 hr at 65°C, treated with RNase A and Proteinase K, and purified on Qiagen PCR purification columns.

For ChIP-qPCR, purified DNA samples were diluted ~10 fold with water and used as templates for qPCR employing primers given in *Supplementary file 4*. Serial dilutions of input DNA served to yield unenriched controls. Fold enrichments were calculated via internal normalization to an

irrelevant negative control locus (in *OTX2*) and subsequent division by corresponding values from input or no-DOX samples. ChIP and input DNA samples were pooled from several independent experiments each, processed using a dedicated Illumina library preparation kit (# IP-202–1012), and sequenced (single-read, 75 cycles) on an Illumina NextSeq 500 system.

BCL files were converted to FASTQ format using Illumina bcl2fastq software. Trimmed reads were mapped to the human genome (hg19) using default settings in Bowtie 2 (*Langmead and Salzberg, 2012*). The Picard suite was subsequently employed for Bam file generation, soft-clipping, and duplicate read marking. ChIP peaks were called using MACS2, based on comparing DOX and input control samples (*Feng et al., 2012*; *Zhang et al., 2008*). Minimum FDR (q value) cutoff for peak detection was 0.01. Heat maps were generated using the plotHeatmap function in deepTools (*Ramírez et al., 2016*). Plotted scores were calculated from normalized bigWig files using the computeMatrix function from the same package. Repeat-free target regions were functionally annotated employing the GREAT analysis suite (*McLean et al., 2010*), using the corresponding hg19 genome coordinates as input (setting: two nearest genes within 1 Mb). Statistically significant hits were subjectively filtered for biological relevance and presented based on the obtained p values. Unbiased motif searches were carried out using the DNA sequences underlying hit regions from *Supplementary file 3* as input to MEME (*Bailey et al., 2009*). ChIP-seq peaks were visualized using custom tracks in the UCSC Genome Browser. ChIP-seq raw data have been submitted to the NCBI GEO repository, under accession number GSE101477.

## Electrophysiological analysis of hESC-CMs

For analysis of hESC-CMs on microelectrode arrays (Multichannel Systems), the electrode areas of plasma-cleaned 9-well MEAs were coated with 3 µl of a 1:75 diluted Matrigel solution in KO-DMEM for approximately 2 hr in a humidified cell culture incubator. Matured CMs were dissociated from maintenance cultures using a 10 x TrypLE Select digestion to obtain a single-cell / small aggregate suspension. Coating solution from the electrode chips was replaced by a ~3 µl suspension droplet containing 25,000–50,000 cells in CM splitting medium. Following fill-up with CM maintenance medium and equilibration over several days, spontaneous beating behavior was recorded and analyzed using MC Rack software.

For action potential (AP) measurements, cultured hESC-CMs were enzymatically dissociated into single cells as previously described (*Meijer van Putten et al., 2015*) and plated at a low density on Matrigel-coated coverslips. AP measurements were performed using the amphotericin-B perforated patch-clamp technique and an Axopatch 200B amplifier (Molecular Devices). Data acquisition and analysis were realized with custom software. Signals were low-pass-filtered with a cutoff of 5 kHz and digitized at 40 kHz. The potentials were corrected for the calculated liquid junction potential of 15 mV (*Barry and Lynch, 1991*). Cell membrane capacitance ($C_m$) was determined with a -5 mV voltage step from $-40$ mV by dividing the time constant of the decay of the capacitive transient by the series resistance.

APs were recorded at $36 \pm 0.2°C$ from single and spontaneously contracting hESC-CMs. Cells were superfused with solution containing (in mM): 140 NaCl, 5.4 KCl, 1.8 $CaCl_2$, 1.0 $MgCl_2$, 5.5 glucose, 5.0 HEPES; pH 7.4 (NaOH). Patch pipettes (borosilicate glass; resistance $\approx 2.5$ M$\Omega$) contained (in mM): 125 K-gluconate, 20 KCl, 5 NaCl, 0.44 amphotericin-B, 10 HEPES; pH 7.2 (KOH). hESC-CMs typically lack the inward rectifier $K^+$ current, $I_{K1}$, that limits the functional availability of $Na^+$ current ($I_{Na}$) and transient outward $K^+$ current ($I_{To}$) (*Giles and Noble, 2016*; *Hoekstra et al., 2012*). To overcome this limitation, we injected an in silico $I_{K1}$ with kinetics of $Kir_{2.1}$ channels through dynamic clamp, as we previously described in detail (*Meijer van Putten et al., 2015*). An amount of 2 pA/pF peak outward current was applied, resulting in quiescent hESC-CMs with a maximal membrane depolarization (MDP) of $-80$ mV or more negative. APs were elicited at 1 Hz by 3 ms, ~1.2 x threshold current pulses through the patch pipette. We analyzed the MDP, maximum AP amplitude ($APA_{max}$), AP duration at 20, 50% and 90% of repolarization ($APD_{20}$, $APD_{50}$, and $APD_{90}$, respectively), maximal upstroke velocity ($V_{max}$) and plateau amplitude ($APA_{plat}$) measured 20 ms after the AP upstroke. Averages were taken from 10 consecutive APs. 4-aminopyridine (4-AP) to block $I_{Kur}$ was used at 50 µM as described previously (*Devalla et al., 2015*).

## Acknowledgements

We thank Dr. Alessandra Moretti for helpful discussions on ISL1 biology, and Dr. Gergana Dobreva as well as Dr. Eva Kutejova for sharing ChIP-seq expertise and data. This work was supported by the Chemical Genomics Centre of the Max Planck Society. RQ thanks the CIM-IMPRS graduate school for support.

## Additional information

### Funding

| Funder | Grant reference number | Author |
| --- | --- | --- |
| CIM-IMPRS graduate school | Graduate student fellowship | Roberto Quaranta |
| Chemical Genomics Centre of the Max Planck Society | CGC II | Boris Greber |

The funders had no role in study design, data collection and interpretation, or the decision to submit the work for publication.

### Author contributions

Roberto Quaranta, Jakob Fell, Arie O Verkerk, Formal analysis, Investigation, Methodology, Writing—review and editing; Frank Rühle, Data curation, Formal analysis, Writing—review and editing; Jyoti Rao, Formal analysis, Investigation; Ilaria Piccini, Data curation, Formal analysis; Marcos J Araúzo-Bravo, Data curation, Software, Formal analysis; Monika Stoll, Resources, Supervision, Funding acquisition; Boris Greber, Conceptualization, Formal analysis, Supervision, Funding acquisition, Investigation, Methodology, Writing—original draft

### Author ORCIDs

Arie O Verkerk http://orcid.org/0000-0003-2140-834X
Boris Greber https://orcid.org/0000-0002-9404-7293

### Decision letter and Author response

Decision letter https://doi.org/10.7554/eLife.31706.027
Author response https://doi.org/10.7554/eLife.31706.028

## Additional files

### Supplementary files

• Supplementary file 1. Genome-wide expression time-series Data denote expression levels [a.u.] of the indicated genes upon cardiac induction of hESCs which were perturbed genetically or signaling-wise: The first two time-series compare ISL1-deficient with *ISL1*-expressing hESCs. The latter two time-courses are based on wild-type hESCs differentiated in the presence or absence of retinoic acid. Both DOX (to induce *ISL1*) and RA were administered at days 3 and 4 such that the pairwise time-series are diverging from day four onwards.
DOI: https://doi.org/10.7554/eLife.31706.017

• Supplementary file 2. Action potential measurements by single-cell patch clamping (A) Baseline AP measurements of four groups of hESC-CM samples: WT and ISL1 KO cells differentiated with or without $RA_{d3,4}$. MDP: maximum membrane depolarization; $APA_{max}$: maximum AP amplitude; $V_{max}$: maximum upstroke velocity; $APD_{20}$, $APD_{50}$, $APD_{90}$: AP duration at 20, 50% and 90% of repolarization, respectively; $APA_{plat}$: plateau amplitude. Conditional formatting highlights the enhanced atrial phenotype of RA-treated ISL1 KO cells. (B) Relative AP parameter values following treatment of the cells with the KCNA5 inhibitor 4-AP (50 μM). Data in each group are expressed as % of untreated controls which were set to 100. Relative changes are highlighted using conditional formatting.
DOI: https://doi.org/10.7554/eLife.31706.018

• Supplementary file 3. Processed ISL1 ChIP-seq data Genomic regions with enriched ISL1 binding as compared to input ChIP DNA. Cells were captured on day 5 of directed cardiac differentiation. Up to two nearest gene loci within 1 Mb are indicated for each hit region. Columns J and M contain expression ratios between ISL1-positive versus ISL1-deficient hESCs at day 5 of differentiation, as extracted from *Supplementary file 1*, to indicate which binding events may directly translate into differential gene expression. Up or downregulated genes are indicated using conditional formatting (green or red shading, respectively).
DOI: https://doi.org/10.7554/eLife.31706.019

• Supplementary file 4. Oligonucleotides (A), antibodies (B), and signaling molecules (C) used in this study.
DOI: https://doi.org/10.7554/eLife.31706.020

• Transparent reporting form
DOI: https://doi.org/10.7554/eLife.31706.021

## Major datasets

The following datasets were generated:

| Author(s) | Year | Dataset title | Dataset URL | Database, license, and accessibility information |
|---|---|---|---|---|
| Quaranta R, Greber B | 2017 | Comparative cardiac induction time-courses using wild-type or genetically modified human embryonic stem cells | https://www.ncbi.nlm.nih.gov/geo/query/acc.cgi?acc=GSE100592 | Publicly available at the NCBI Gene Expression Omnibus (accession no: GSE100592) |
| Roberto Quaranta, Frank Rühle, Monika Stoll, Boris Greber | 2017 | ISL1-binding sites in human ES cell-derived cardiomyocyte progenitor cells identified by ChIP-seq upon cardiac differentiation | https://www.ncbi.nlm.nih.gov/geo/query/acc.cgi?acc=GSE101477 | Publicly available at the NCBI Gene Expression Omnibus (accession no: GSE101477) |

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
