## [Decision Letter]

Thank you for submitting your article "Revised Roles of ISL1 in a hES Cell-Based Model of Human Heart Chamber Specification" for consideration by *eLife*. Your article has been favorably evaluated by K VijayRaghavan (Senior Editor) and three reviewers, one of whom is a member of our Board of Reviewing Editors. The following individual involved in review of your submission has agreed to reveal his identity: Joshua Waxman (Reviewer #3).

The reviewers have discussed the reviews with one another and the Reviewing Editor has drafted this decision to help you prepare a revised submission.

Summary:

Quaranta et al. reinvestigated the function of the LIM transcription factor ISL1 (ISLET-1) during cardiac differentiation using human embryonic stem cells (hESCs) as a model of early cardiogenesis. To investigate the function of ISL1, an ISL1 knockout hESC line was generated. Using a novel differentiation protocol of hESCs, the authors observe that depletion of ISL1 slows down the differentiation process. This phenotype is reverted when induction of ISL1 transgene is made on days 3-4. Based on these results they conclude that ISL1 is not required to stabilize the cardiac precursor cell state but rather accelerates cardiomyocyte formation, defining ISL1 as a pro-differentiation factor. To validate the hypothesize that the ISL1-depletion causes a switch in cardiomyocyte subtype identity, a series of experiments, including microarray analysis and competing induction by retinoic acid (RA), were performed. Therefore, they propose that ISL1 promotes ventricular cardiomyocyte specification and counteract an atrial fate induced by RA. in vivo experiments and functional genomics analysis show that ISL1 functionally antagonizes atria specification driven by RA signaling. A regulatory module of this competition involving repression of Isl1 and activation of Nk2f1 by MEIS1, and repression of Nk2f1 by ISL1, is proposed in conclusion.

The work is carefully and thoughtfully done and directed by strategic analyses. In general, this study is an interesting investigation because ISL1 is a crucial factor for heart development. Although it shows alignment with in vivo analysis of the role of RA in SHF development and has the merit to revise the view of ISL1 function during cardiac differentiation in pluripotent stem cells, there remains discrepancy with previous works. Indeed, overexpression of Isl1 in mouse ESCs results in normally beating cardiomyocytes, highly enriched in the atrial subtype at the expense of the ventricular lineage. Hence, this discordance should be addressed more critically. It is important to know if data are exclusive to hESCs or can also be applied to in vivo context. Furthermore, for this to be a compelling contribution, new regulatory links established in this study between MEIS1 and Isl1, and between ISL1 and Nr2f1, should be substantially strengthened.

Essential revisions:

1) Subsection “ISL1 accelerates pan-cardiac gene induction in hESCs”, last paragraph. Authors should include GO term analysis for the 73 cardiac and uncharacterised genes that are differentially expressed between WT and Dox-induced ISL1 in the KO background. The authors' claim for accelerated CM formation needs stronger justification.

2) Subsection “ISL1 KO phenocopies an atrial wild-type CM phenotype induced by retinoic acid”, last paragraph. The effects on action potentials after inhibition of KCNA5 seem weak. Would this be expected from knowledge of the involvement of this channel in atrial action potentials?

3) How does the ISL1 ChIP data compare to other data sets in target cell, size and overlap? The low number of apparent targets may be a technical and therefore a conceptual weakness. How can this be strengthened?

4) What is the outcome on MEIS2 after over-expression of ISL1. This important link seems to be missing. The authors suggest a unidirectional network. But perhaps ISL1 can also inhibit MEIS2. There would be many examples in the literature of segregation of cell fates through mutual TF antagonism (e.g. cardiac vs. blood).

5) The work of Mohun and antagonism between NKX2-5 and MEIS TFs should be considered and cited. How does it sit with this model? An important point for discussion as the ISL1 binding site revealed by ChIP in the current study is very similar to the NKE.

6) The regulatory relationships between ISL1 and Nr2f1 and between MEIS1 and Nr2f1 should be functionally tested using gain and loss of function and better analysis of ChIP and putative regulatory elements. The authors performed ISL1 ChIP-seq analysis to identify direct target genes. However, the choice of the ISL1 enhancers is not totally explained. For example, why the NRF2 (-476kb) or MEF2C (-225kb) were selected? The ISL1 bound region in Nr2f1 is ~0.5Mb away from the gene body and no functional data is provided to suggest it acts as a distal repressor. A more detailed analysis should be made. It is important to know whether other cardiac TF-binding sites are found around this region. Enhancer function should be tested in vivo or in vitro.

7) The authors used a published protocol for hESC differentiation, however characterization of cardiomyocytes origin (FHF vs. SHF) is missing in this study and potentially bears on how the current data meshes with in vivo studies on RA function in the SHF.

8) It is surprising to observe that peak of ISL1 expression, at day 5, corresponds to the activation of differentiated markers (cTnT, cTnC…). Please comment.

9) The time-course of ISL1 overexpression is not fully analyzed. It will be important to assess whether its overexpression affects the fate of multipotent cardiovascular progenitors at an earlier stage. Indeed, induction of ISL1 on days 3-4 is not sufficient to understand its function.

10) Proliferation of cardiac progenitors after ISL1 expression and deletion should be correctly addressed. Gene expression here is being used only to define identity, but could also reflect flux in cell populations.

11) The authors observe that ISL1 suppresses the atrial induction by RA signaling. However initially it was shown that late induction of ISL1 (3 weeks) antagonises the expression of ventricular specific gene Mlc2v (Figure 2). The timing of induction of ISL1 expression in all subsequent experiments should be clarified and it should be indicated how early induction relates to atrial inhibition. For example, in Figure 6, Dox induction of ISL1 is done at d3 and d4.

12) Functional activity of differentiated cardiomyocytes was used to identify the subtype identity of these cells. However large images of differentiated cardiomyocytes should be provided.

13) Antagonism between RA signaling and ISL1 was mainly addressed through RA activation. It is important to examine the effect of RA deficiency in this context.

14) The choice of MEIS2 should be better explained.

15) Their observation that ISL1 accelerates CM differentiation in hES-derived cardiomyocytes is pretty much exactly what would be expected. From extensive in vivo and in vitro work, ISL1 is a marker of second heart field progenitors whose transient expression is required for cardiomyocyte differentiation. Thus, one would not really expect a primary requirement in the amplification of the progenitor population. In fact, signals that promote the proliferation of this population, such as canonical Wnt signaling, inhibit ISL1 expression and cardiomyocyte differentiation. This should be better discussed.

16) While it's true that ISL1 KO mice still have some atrial cells, these are thought to be first heart field-derived. More importantly, the ISL1 KO mice have significant atrial defects, arguing against their conclusion. Furthermore, in zebrafish, ISL1 is required for atrial differentiation at the venous pole. Please discuss.

17) Loss of Meis factors only tempers the induction ability of retinoic acid, suggesting RA must induce/repress other critical factors. Please comment.

18) Whether NR2F1 is required for the atrial differentiation effect observed in ISL1 KO cells should be tested. It is not clear that Nr2f1 can be accepted as a master regulator of atrial differentiation, at least by itself in hES cells, and the major factor driving cardiomyocyte specialization in this context.

---

## [Author Response]

Essential revisions:1) Subsection “ISL1 accelerates pan-cardiac gene induction in hESCs”, last paragraph. Authors should include GO term analysis for the 73 cardiac and uncharacterised genes that are differentially expressed between WT and Dox-induced ISL1 in the KO background. The authors' claim for accelerated CM formation needs stronger justification.

We are happy to provide a corresponding GO analysis in new Figure 1—figure supplement 1. The gene set is highly enriched for cardiac myocyte terms, thereby strengthening our interpretation of ISL1 being an accelerator of cardiac differentiation. As detailed below, we have also carried out experiments to rule out alternative explanations, notably positive effects of ISL1 on precursor cell proliferation (new Figure 1). Together with the previous data, we think the differentiation-accelerating function of ISL1 is now well documented and analyzed. We thank the reviewer for encouraging these additional efforts.

2) Subsection “ISL1 KO phenocopies an atrial wild-type CM phenotype induced by retinoic acid”, last paragraph. The effects on action potentials after inhibition of KCNA5 seem weak. Would this be expected from knowledge of the involvement of this channel in atrial action potentials?

In the present study, we observed clear effects of IKur blockade on the RA- and ISL1-KO action potentials, especially in the early repolarization phase. This in agreement with previous findings in RA-treated HES-CMs and hiPSC-CMs, where IKur was inhibited by blockers [Devalla et al., PMID 25700171] or by genetic manipulation [Marczenke et al., PMID 28729840]. It is also largely similar as found in freshly isolated human atrial myocytes [Wang et al., PMID 8222078; Ford et al., PMID 23364608; Ford et al., PMID 26455450]. The effects of IKur blockade are most pronounced in the early repolarization phase of atrial action potentials, because IKur is active at these potentials (-30 mV and more positive) [Wang et al., PMID 8222078]. Effects on the final repolarization phase are less clear, and can differ between studies. This is because IKur inhibition raises the plateau level to more positive voltages and consequently there is an increase of the repolarizing rapid delayed rectifier potassium current. This may counteract the action potential prolonging effect of IKur inhibition [for review, see Nattel et al., PMID 10575199]. We are aware that the IKur blockade-induced elevation of the action potential plateau seems more pronounced in freshly isolated human atrial myocytes [Ford et al., PMID 26455450] than in hES-CMs and hiPSC-CMs. The exact reason it unknown. We noticed stronger effects in our previous hiPSC-based study [Marczenke et al., PMID 28729840], so it might be an issue with the HuES6 background used here throughout. Furthermore, developmental changes in IKur have been described with a larger contribution to repolarization in adult state [Trépanier-Boulay et al., PMID 15364616]. The overall role of IKur in atrial hPSC action potential repolarization may thus be smaller than in freshly isolated human atrial myocytes.

3) How does the ISL1 ChIP data compare to other data sets in target cell, size and overlap? The low number of apparent targets may be a technical and therefore a conceptual weakness. How can this be strengthened?

As acknowledged in the first paragraph of the subsection “ISL1 functionally antagonizes atrial specification driven by RA signaling”, technical reasons for the rather low fold enrichments cannot be fully excluded. Through personal communication with ISL1 experts, we have learned that there is virtually no commercially available ISL1 antibody that would be highly suited for ChIP, and our protocol tends to give better enrichments in case of other targets. To our knowledge, our data set represents the first ISL1 ChIP-seq analysis in a cardiac context. (There is no published mouse one, for instance, to compare our data set with.) Moreover, many target genes showed differential gene expression when comparing ISL1+ and ISL1-deficient cells, while ISL1 KO cells served as an excellent specificity control in the ChIP-seq analysis. Furthermore, the set of target genes was enriched for meaningful annotation terms and the identified motif resembles previously identified ones (Mazzoni et al., PMID 23872598). Finally, with regards to the overlap with the differential expression data (Figure 6), we also wish to point out that the effects of ISL1 disruption / induction on the cells' transcriptome are not really massive at an appropriate early time point (d 5). Hence, we think the rather small size of the ChIP-seq data could simply reflect the reality rather than presenting a "conceptual weakness", so we would like to stick to our interpretation.

4) What is the outcome on MEIS2 after over-expression of ISL1. This important link seems to be missing. The authors suggest a unidirectional network. But perhaps ISL1 can also inhibit MEIS2. There would be many examples in the literature of segregation of cell fates through mutual TF antagonism (e.g. cardiac vs. blood).

A valid and interesting point, also because *MEIS2* is part of the ChIP-seq set of genes (Supplementary file 3). New data generated are in Figure 5—figure supplement 1: There is a very modest effect of ISL1 on *MEIS2*. Since the effect was not statistically significant, we did not incorporate it into our model. So, the ISL1 KO phenotype and the rescue by pulsed ISL1 induction appears to be MEIS2-independent. In this particular case, gene regulation appears to be unidirectional, with MEIS2 being able to downregulate ISL1 and not vice versa.

5) The work of Mohun and antagonism between NKX2-5 and MEIS TFs should be considered and cited. How does it sit with this model? An important point for discussion as the ISL1 binding site revealed by ChIP in the current study is very similar to the NKE.

We are happy to cite the paper by Dupays et al. in our Discussion (subsection “Antagonism with RA signaling and cardiac subtype-specification module”, first paragraph) as it provides strong support for a functional role of MEIS1/2 in cardiac development. Although our study is ISL1-centred, in the revised manuscript we now also provide an analysis of terminally differentiated MEIS1/2 DKO cells that showed a mild but interesting phenotype (positive staining for CM markers but no spontaneous beating – Figure 4—figure supplement 1). An extended discussion of MEIS2 would certainly be justified if there was a mutual regulation between *ISL1* and *MEIS2*. Since it is mostly unidirectional, though (see previous point and new Figure 5—figure supplement 1), we would rather want to restrict our discussion to MEIS2 as a mediator of the ISL1-repressing effect by RA.

6) The regulatory relationships between ISL1 and Nr2f1 and between MEIS1 and Nr2f1 should be functionally tested using gain and loss of function and better analysis of ChIP and putative regulatory elements. The authors performed ISL1 ChIP-seq analysis to identify direct target genes. However, the choice of the ISL1 enhancers is not totally explained. For example, why the NRF2 (-476kb) or MEF2C (-225kb) were selected? The ISL1 bound region in Nr2f1 is ~0.5Mb away from the gene body and no functional data is provided to suggest it acts as a distal repressor. A more detailed analysis should be made. It is important to know whether other cardiac TF-binding sites are found around this region. Enhancer function should be tested in vivo or in vitro.

As to MEIS2, we think the data in Figure 4 and Figure 4—figure supplement 1 provides strong evidence for the repressive effect on *ISL1*. Based on searches for a published MEIS1 motif in the *ISL1* locus, we have now carried out ChIP-qPCR analyses of various candidate regions (without being able to validate the antibody). None of the >15 tested regions gave a significant enrichment, so we are not including these data, while acknowledging that the uncovered regulatory link between *MEIS2* and *ISL1* is based on clear-cut transcriptional effects: Importantly, this part of the overall model is supported both by gain and loss-of-function experimentation using MEIS2 TET-ON as well as MEIS1/2 double-knockout cells (Figure 4, respectively). With regards to the DKO analysis, we felt it was important to validate that these mutant cells are still capable of forming cardiac-like cells, which we now provide as new Figure 4—figure supplement 1.

The presentation and qPCR confirmation of selected ISL1 targets regions was based on the unbiased analysis of the ChIP-seq data set. As already mentioned, we sought to use the most stringent assay for evaluating the functionality of the -476kb element upstream of *NR2F1*, namely, to delete this region on an ISL1^KO/I.TET-ON^ background. The ultimate experiment was then to ask whether in this -476kb mutant line, *ISL1* induction on ISL1 KO background would fail to downregulate *NR2F1* at the cardiac precursor stage. The experiment is shown in Figure 6—figure supplement 1 and regrettably, it did not reveal the predicted defect. Hence, we have toned down on our interpretation (subsection “ISL1 functionally antagonizes atrial specification driven by RA signaling”, second paragraph) although transcriptionally, the effect is highly reproducible and strong. This fact is also underscored by additional data generated during this revision. For instance, Figure 5 shows that endogenous RA signaling is not involved in *NR2F1* upregulation in the *ISL1* knockout, and Figure 3—figure supplement 1 highlights the cooperativity of ISL1 KO and RA signaling in promoting *NR2F1* induction. We hope the reviewer acknowledges our attempts to strengthen the data.

7) The authors used a published protocol for hESC differentiation, however characterization of cardiomyocytes origin (FHF vs. SHF) is missing in this study and potentially bears on how the current data meshes with in vivo studies on RA function in the SHF.

We agree it is an interesting question which developmental route hESCs actually take for acquiring a cardiomyocyte identity. We are devoting an entire paragraph to discussing this matter (subsection “Paradigm for second heart field development?”). As stated in this section, the difficulty in assessing the point resides in the lack of definite markers distinguishing between FHF and SHF. ISL1 itself is the prototype marker for the SHF (Cai et al., PMID 14667410), yet Dorn et al. (PMID 25524439) have recently shown that *Isl1* is also expressed in the FHF – albeit very transiently and likely without playing any important role there. So, the issue needs to be treated with caution and accordingly, we are discussing it in a rather careful manner. Notably, in our protocol, *ISL1* expression is very prominent and lasts for several days. Moreover, the cells are – like the SHF – functionally responsive to RA to become fated into atrial CMs. Hence, we believe this present study itself, with its thorough analysis of ISL1 function and its integration with RA signaling, may serve as an indication that the cells take a SHF route upon directed differentiation. At least our model fits best to this idea. It would be interesting to investigate if alternative protocols – notably BMP-driven ones – yield different *ISL1* expression kinetics or significantly lower expression levels.

8) It is surprising to observe that peak of ISL1 expression, at day 5, corresponds to the activation of differentiated markers (cTnT, cTnC…). Please comment.

Indeed, and it is for this reason why we interpret these rapid kinetics as ISL1 being a functional driver of terminal cardiac differentiation. In our protocol, ISL1 expression commences at day 3 and reaches near-maximum levels as early as by day 4. In revised Figure 1, we now show the kinetics of *ISL1* induction at increased temporal resolution. This way, it can now be better appreciated that *ISL1* precedes the upregulation of terminal differentiation genes and conversely, their induction kinetics becomes delayed in an ISL1-deficient scenario. Interestingly, though, our time course and ChIP-seq analyses suggests that ISL1 may not directly activate many structural CM genes (with few exceptions like *MYLK3* – l. 356, Supplementary file 3) but may rather control additional players like *MEF2C* and *NKX2.X* genes. These immediate-early targets show an even earlier induction compared with structural CM genes, i.e. they start becoming expressed approximately one day earlier (see day 4 data of *MEF2C* in Figure 1/H, NKX2.5 data in Figure 1, and Figure 1—figure supplement 1). As stated at the end of the Discussion section, we think our study certainly encourages the functional investigation of additional players involved in the cardiac program.

9) The time-course of ISL1 overexpression is not fully analyzed. It will be important to assess whether its overexpression affects the fate of multipotent cardiovascular progenitors at an earlier stage. Indeed, induction of ISL1 on days 3-4 is not sufficient to understand its function.

We are not sure whether we can agree to this point. Overall, given the rapid course of events in cardiac differentiation, we felt careful time course analyses would present the best means of analyzing the downstream effects of ISL1. In this regard, the TET-ON system proved to be a great tool allowing us to compare ISL1+ and ISL- scenarios in otherwise identical cell populations. (That is, any potential technical / cell batch etc. variability between e.g. WT and KO cells becomes eliminated this way.) We used DOX induction at days 3 and 4 in most experiments, as this most closely mimics the transient expression of endogenous ISL1 in WT cells. Revised Figure 1 now shows this pattern at increased temporal resolution. Indeed, pulsed *ISL1* induction nicely rescued all phenotypes of ISL1 KO cells and our genome-wide time course analysis strongly supports this notion (Figure 1—figure supplement 1, Supplementary file 1). Nonetheless, we have also investigated the consequences of continuous ISL1 overexpression up until the early CM stage, which is shown in Figure 2: There was no indication that ISL1 would arrest the differentiation process in any way. In further support of this notion, we now also show that ISL1 does not influence the expression levels of other precursor markers (new Figure 2—figure supplement 1) and that it does not significantly influence cell proliferation (new Figure 1). While we do acknowledge that long-term (i.e. non-physiological) ISL1 overexpression interferes with ventricular subtype specification as observed by Dorn et al. using mouse cells (PMID 25524439), we think our short-term analyses provide compelling evidence for ISL1 acting as an accelerator of cardiac differentiation.

10) Proliferation of cardiac progenitors after ISL1 expression and deletion should be correctly addressed. Gene expression here is being used only to define identity, but could also reflect flux in cell populations.

We agree and have carried out cell counting combined with cell cycle analysis based on propidium iodide staining (new Figure 1). There were no significant differences between WT and ISL1 KO cells. While these results do not fully argue against a role of ISL1 in cell proliferation in general, they demonstrate that, in context of our directed differentiation protocol, the differences in cardiac induction kinetics are not due to an altered proliferation ability of ISL1 KO cells. We thank the reviewer for bringing up this important aspect.

11) The authors observe that ISL1 suppresses the atrial induction by RA signaling. However initially it was shown that late induction of ISL1 (3 weeks) antagonises the expression of ventricular specific gene Mlc2v (Figure 2). The timing of induction of ISL1 expression in all subsequent experiments should be clarified and it should be indicated how early induction relates to atrial inhibition. For example, in Figure 6, Dox induction of ISL1 is done at d3 and d4.

DOX at days 3 and 4 was consistently used throughout unless stated otherwise (also see response to point 9 above). This fact is now clearly stated: "Unless stated otherwise, the endogenous *ISL1* expression pattern was mimicked by pulsed DOX administration on days 3 and 4 using ISL1^KO/I.TET-ON^ cells." In addition, DOX timing is indicated in virtually all figures or accompanying legends (e.g. as "DOX_d3,4_"). In Figure 3, where this fact admittedly may not have been fully clear, we have added the following statement to the legend: "In case of using ISL1^KO/I.TET-ON^ cells, all ISL1^+^ data in this figure are based on a day 3-4 treatment with DOX". In some cases throughout this study, we used continuous long-term or late-stage DOX treatment like for instance in new Figure 2, which is indicated in the treatment scheme in that same figure. Beyond that, the exact ISL1 overexpression duration in the short-term did not actually appear to be too critical: As shown in Figure 2, continuous *ISL1* overexpression from day 3-7 did by no means interfere with terminal cardiac differentiation but accelerated the process – just as a day 3-4 treatment did (Figure 1 and Figure 1—figure supplement 1). Interestingly, the time window of maximum RA responsiveness (d 3-4) matched well with the time window of endogenous *ISL1* induction (see optimization data for RA-mediated atrial induction in Figure 3—figure supplement 1) and we think this fact relates to the elucidated functional antagonism between RA and ISL1 as reflected by our model.

12) Functional activity of differentiated cardiomyocytes was used to identify the subtype identity of these cells. However large images of differentiated cardiomyocytes should be provided.

We can agree to showing the pictures at higher magnification since we already show bulk analysis of the same kinds of samples in the previous panel (Figure 3) using immunoblotting. As requested, in revised Figure 3, the specificity of the stainings can now be better appreciated – some subcellular structure in case of MLC2v as typical for the still rather early time point of analysis (3 wk), and a perinuclear / speckled subcellular pattern for the secreted factor ANP.

13) Antagonism between RA signaling and ISL1 was mainly addressed through RA activation. It is important to examine the effect of RA deficiency in this context.

A good point which is also relevant in light of a recent study by the Keller laboratory (PMID 28777944). In order to examine RA deficiency, we first validated and then employed the pan-RA receptor antagonist AGN 193109 (AGN) (Agarwal et al., PMID 8647816). Hence, in new Figure 5—figure supplement 1, we first show that AGN strongly diminishes the RA-driven induction of the early atrial specifier *NR2F1* (and *NR2F2*) in a dose-dependent fashion. In the actual experiment, AGN treatment then did not rescue the phenotype of ISL1 KO cells. In particular, it did not lead to the repression of *NR2F1* (new Figure 5). Hence, these results importantly demonstrate that ISL1 suppresses *NR2F1* through a RA-independent regulatory axis. We thank the reviewer for the interesting suggestion.

Surprisingly, AGN treatment in ISL1 KO cells actually lead to a further increase of *NR2F1* (qPCR and Western blot) and of the independent atrial marker *DHRS9* (qPCR). This outcome may be explained by the fact that in absence of RA, the RA receptors exert a repressive effect on their target genes (reviewed by Cunningham and Duester, PMID 25560970; Nagy et al., PMID 9150137). In this case, the inhibition of the receptors mediated by AGN could act by abolishing this repressive effect in the absence of RA, leading to increased gene expression of the targets.

14) The choice of MEIS2 should be better explained.

This information is now provided in the legend to Figure 4: We used a 3-fold expression difference between the +RA and -RA samples at day 5 as a first filtering criterion, while additionally requiring a pronounced immediate-early effect on gene expression in the range of hours.

15) Their observation that ISL1 accelerates CM differentiation in hES-derived cardiomyocytes is pretty much exactly what would be expected. From extensive in vivo and in vitro work, ISL1 is a marker of second heart field progenitors whose transient expression is required for cardiomyocyte differentiation. Thus, one would not really expect a primary requirement in the amplification of the progenitor population. In fact, signals that promote the proliferation of this population, such as canonical Wnt signaling, inhibit ISL1 expression and cardiomyocyte differentiation. This should be better discussed.

This is an insightful comment and we tend to fully agree to this view – albeit mostly after having carried out this present study. In the literature, we failed to find hardly any evidence for a pro-differentiation function of ISL1 but rather, it is commonly used as *the* gene marking the progenitor state implying a functional role for stabilizing it. This is reviewed in our Introduction and in the beginning of our Discussion (Cao et al., PMID 23896987; Cohen et al., PMID 17607356; Qyang et al., PMID 18371348; Zhang et al., PMID 26942852). These studies also suggested a positive role of WNT signaling to promote ISL1 and cardiac precursor identity. It is for this reason that we also investigated various signaling molecules, including previously published cocktails, for their ability to stabilize the transient ISL1^+^ cell state and that we carefully examined whether sustained *ISL1* expression would arrest cardiac differentiation (Figure 2 and Figure 2—figure supplement 1). To our knowledge, there is only one study that explicitly questions any positive role of WNT signaling in promoting cardiac precursor identity and of ISL1 being a stabilizer this stage (Kwon et al., PMID 19620969). We are citing this work in the last paragraph of the subsection “Differentiation-promoting function of ISL1”. In addition, we have added a sentence in the Discussion to underscore the controversial views in the field: "In support of this view, Kwon et al. (2009) proposed that ISL1 may actually be counterproductive for sustaining the cardiac precursor state." After all, our study presents the first investigation of ISL1 function in the human system and given the conflicting views in the field, we hope this work will help to clarify the functional roles of this key factor – beyond its differentiation-accelerating effect highlighted in Figure 1 and 2.

16) While it's true that ISL1 KO mice still have some atrial cells, these are thought to be first heart field-derived. More importantly, the ISL1 KO mice have significant atrial defects, arguing against their conclusion. Furthermore, in zebrafish, ISL1 is required for atrial differentiation at the venous pole. Please discuss.

Upon carrying out the present study we have learned about a revised view of SHF development that says the vast majority of atrial cells, too, are SHF-derived (e.g. in Burridge and Wu, PMID 26631515; Zaffran et al., doi:10.3390/jdb2010050). As stated in the Introduction (first paragraph), this paradigm change was apparently based on improved *Isl1*-Cre driver lines that were introduced several years after establishing ISL1 as a pan-SHF marker (Cai et al., PMID 14667410; Yang et al., PMID 16556916). At about E7.5, however, retinoic acid kicks in to act on the posterior part of the SHF where it promotes atrial specification while confining *Isl1* expression to the anterior portion (Hochgreb et al., PMID 13129847; Sirbu et al., PMID 18498088; Ryckebusch et al., PMID 18287057). Hence, despite initially marking the entire (early) SHF, ISL1 may ultimately be incompatible with atrial specification and our competition experiments show that this is indeed the case. Admittedly, the situation that the early SHF is all ISL1^+^ whereas it later becomes divided into an ISL1^-^ (posterior) and ISL1^+^ (anterior) domain is a bit confusing and may impede the understanding of ISL1 function. It is for this reason that we think the hESC system can make a significant contribution here.

Overall, our data highlight a positive role of ISL1 in ventricular specification – in line with its prominent expression in the anterior SHF – and a negative one for atrial induction, in agreement with the exclusion of *Isl1* from the posterior SHF. Hence, we do think our model actually agrees well with the mouse *Isl1* knockout phenotype. Notably, our findings are related to the dynamic subtype specification process, not to preceding events. For instance, atrial defects seen in *Isl1* knockout mice could also be due to a reduced number of early pan-SHF progenitors or due to an impaired migration ability of these cells. Although our study is exclusively based on the hESC system, we extensively (and carefully) discuss our findings in light of the in vivo context (subsection “Paradigm for second heart field development?”), which has now been extended by clarifying a seeming discrepancy with the Dorn et al. study (PMID 25514439) – subsection “Antagonism with RA signaling and cardiac subtype-specification module”, third paragraph. In sum, we attempt to incorporate current views of heart chamber development to the best of our knowledge, whereas in principle, we would not deny that there might also be alternative ways of interpreting our data.

17) Loss of Meis factors only tempers the induction ability of retinoic acid, suggesting RA must induce/repress other critical factors. Please comment.

We would agree. In the *MEIS1/2* double knockout, the ISL1 target *MEF2C* does not reach normal expression levels under RA, for instance, suggesting that there may be additional mechanisms downstream of RA leading to functional *ISL1* depletion and release of the atrial program (Figure 4). The model of Figure 6 is meant to best reflect our findings in terms of the two competing branches converging of *NR2F1*. Certainly, though, RA does more than merely triggering *MEIS2* and *NR2F1* expression and hence, the disclosure of the complete +/- RA expression time courses in Supplementary file 1 is also meant to serve as a transparent starting point for further investigation.

18) Whether NR2F1 is required for the atrial differentiation effect observed in ISL1 KO cells should be tested. It is not clear that Nr2f1 can be accepted as a master regulator of atrial differentiation, at least by itself in hES cells, and the major factor driving cardiomyocyte specialization in this context.

As indicated in the illustration of Figure 6, the importance of *NR2F1* downstream of RA is based on the revealing work by Devalla et al. (PMID 25700171) in hESCs showing that it controls several key atrial ion channel genes like *KCNA5*. In full agreement with this, our study highlights the importance of *ISL1* suppression to release its repressive block on *NR2F1*. We do not deny that there may be additional mechanisms and factors involved. Moreover, in an effort meant to be complementary to the Devalla et al. study, we successfully knocked out *NR2F1* in our ISL^KO/I.TET-ON^ line. We then sought to ask whether this defect would impede atrial induction as promoted by the *ISL1* knockout. As already mentioned, though, following a total of three rounds of genetic manipulation (ISL1 KO + ISL1 TET-ON + NR2F1 KO), these cells failed to differentiate properly in our hands, which unfortunately prevented us from incorporating these significant efforts into the revised manuscript. We are aware, therefore, that we are lacking the ultimate proof at this point but hope the reviewer will acknowledge that we tried hard in extending our findings.